# TRANSFORMERS CAN DO BAYESIAN CLUSTERING

## ABSTRACT

Bayesian clustering accounts for uncertainty but is computationally demanding at scale. Furthermore, real-world datasets often contain missing values, and simple imputation ignores the associated uncertainty, resulting in suboptimal results. We present Cluster-PFN, a Transformer-based model that extends Prior-Data Fitted Networks (PFNs) to unsupervised Bayesian clustering. Trained entirely on synthetic datasets generated from a finite Gaussian Mixture Model (GMM) prior, Cluster-PFN learns to estimate the posterior distribution over both the number of clusters and the cluster assignments. Our method estimates the number of clusters more accurately than handcrafted model selection procedures such as AIC, BIC and Variational Inference (VI), and achieves clustering quality competitive with VI while being orders of magnitude faster. Cluster-PFN can be trained on complex priors that include missing data, outperforming imputation-based baselines on real-world genomic datasets, at high missingness. These results show that the Cluster-PFN can provide scalable and flexible Bayesian clustering.

## 1 INTRODUCTION

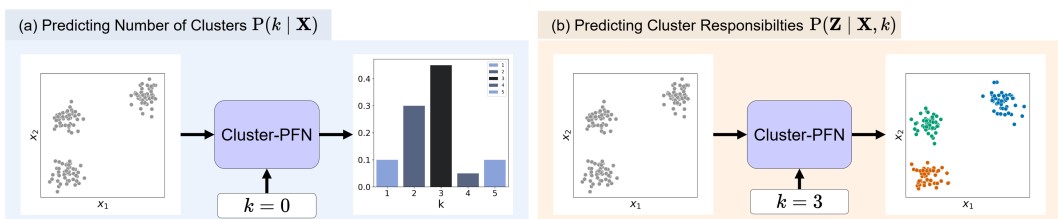

Figure 1: Cluster-PFN usage. In (a), we provide $k = 0$ to the Cluster-PFN, which signals the Cluster-PFN it should predict the number of clusters $P(k|X)$. In (b), we provide $k = 3$ to the Cluster-PFN, meaning it should group the data into 3 clusters. In this case, it will estimate the probability for each cluster for each object, also called the cluster responsibilities.

Clustering is an unsupervised learning technique that aims to group similar data points together. In this work, we focus on model-based clustering, where a probabilistic model, in our case a finite Gaussian Mixture Model (GMM), is used to represent the data. However, GMMs rely on point estimates of parameters and do not account for uncertainty.

Bayesian approaches address this limitation by placing prior distributions over model parameters, allowing for more robust and uncertainty-aware clustering. While this provides a principled way to incorporate uncertainty into the clustering process, inference in Bayesian models is typically computationally intensive, even when using approximate inference methods such as Variational Inference (VI) (Hoffman et al., 2013; Jordan et al., 1999; Wainwright & Jordan, 2008).

Müller et al. (2024) introduced Prior-Data Fitted Networks (PFNs), which are meta-learned models that approximate Bayesian inference. Built on the Transformer architecture (Vaswani et al., 2017), PFNs learn to approximate posterior probabilities in supervised settings using a single forward pass, trained entirely on synthetic data generated from a prior. PFNs have demonstrated improved speed and accuracy compared to traditional Bayesian inference methods (Müller et al., 2024; Adriaensen et al., 2023).

We ask whether a PFN can be trained to perform Bayesian clustering by learning from synthetic data generated under a GMM clustering prior. We propose Cluster-PFN, a Transformer-based model that extends the PFN framework to the unsupervised setting, enabling efficient, accurate, and flexible Bayesian clustering. See Figure 1 for its usage: it can estimate the posterior over the number of clusters, and it can estimate cluster responsibilities. We structure our investigation around the following research questions:

**RQ1: Can the Cluster-PFN effectively cluster synthetic and real-world datasets?** We show that Cluster-PFN can cluster data with up to five features, provide posterior probabilities for cluster memberships, and can provide a user-specified number of clusters.

**RQ2: How does the Cluster-PFN compare to GMMs with handcrafted criteria (AIC, BIC, silhouette) and VI in estimating the number of clusters?** We find that Cluster-PFN predicts the number of clusters with greater accuracy than handcrafted criteria and VI. Our Cluster-PFN is the first model that can learn approximations to estimate the number of clusters in a scalable manner, and by a similar argument of Müller et al. (2024), our Cluster-PFN approximates the posterior.

**RQ3: How does the Cluster-PFN perform against VI in terms of clustering quality and uncertainty estimation?** We demonstrate that Cluster-PFN performs similarly to VI across external metrics, even when evaluated on datasets much larger than those used for training, while being orders of magnitude faster.

**RQ4: How does the Cluster-PFN model compare to baseline imputation methods in terms of external evaluation metrics when handling missing data?** While VI requires particular prior distributions, the PFN can learn from arbitrary priors. We illustrate this by training a PFN on a prior that integrates missing data (at random). We show that for real world genomic data, the Cluster-PFN outperforms baseline models at missingness levels of 30% and above.

Section 2 outlines the background of Bayesian inference and PFNs. Section 3 describes the Cluster-PFN. Section 4 describes the experimental setup and Section 5 presents the results. Additional results are provided in the appendices, including different priors, experimental conditions, additional metrics, and comparisons with more baselines. Section 6 discusses the approximations made by Cluster-PFN and broader implications. Section 7 concludes with a summary and future directions.

## 2 BACKGROUND

In this section, we discuss Bayesian Clustering and Prior-Data Fitted Networks (PFNs).

**Bayesian Clustering.** In the context of clustering, we consider a dataset of observations $\boldsymbol{X} = \{x_1, x_2, \ldots, x_n\} \subset \mathbb{R}^d$ where each observed data point $x_i$ is associated with a latent discrete variable $z_i$ indicating its cluster assignment. Furthermore, there is a probabilistic model that models the density of each cluster; that is, the GMM model. Say $z_i = 1$, then the first Gaussian Mixture determines the density of this $x_i$. Assuming $K$ clusters, the GMM model has $K$ means and $K$ covariance matrices. We refer to these parameters as $\theta$. The goal is to infer the cluster assignments (also called responsibilities) $\mathbf{z} = z_1, \ldots, z_n$, along with estimates of the cluster parameters.

In the Bayesian setting, the aim is not only to get estimates of the parameters, but also to model their uncertainty. These uncertainties are naturally also incorporated into the responsibilities. Bayesian approaches incorporate uncertainty by placing a prior over latent variables and model parameters, and then inferring a posterior distribution over possible clusterings given the observed data. Using Bayes' theorem, the posterior can be expressed as:

$$p(\theta, \mathbf{z} \mid \boldsymbol{X}) = \frac{p(\theta, \mathbf{z})p(\boldsymbol{X} \mid \theta, \mathbf{z})}{p(\boldsymbol{X})} = \frac{p(\theta, \mathbf{z})p(\boldsymbol{X} \mid \theta, \mathbf{z})}{\int_{z,\theta} p(\theta, \mathbf{z})p(\boldsymbol{X} \mid \theta, \mathbf{z})dzd\theta} \tag{1}$$

However, exact inference of this posterior is generally intractable due to the difficulty of marginalizing over all latent variables and parameters when computing the marginal $p(\mathbf{X})$. This is a high-dimensional integral over $z$ and $\theta$.

To make inference tractable, several algorithms can be used, such as Variational Inference (VI) and Markov-Chain Monte Carlo (MCMC) (Andrieu et al., 2003; Neal, 2012; Welling & Teh, 2011). We

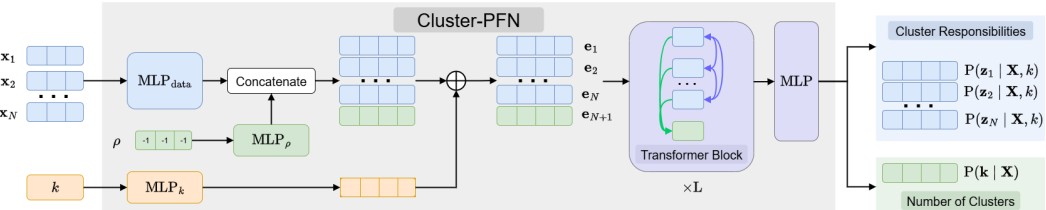

Figure 2: Full training pipeline of the Cluster-PFN

focus on VI, which is commonly used for Bayesian clustering with GMMs (when using conjugate priors (Bishop, 2006, p. 474)). VI approximates the posterior with a simpler, factorized distribution: $q(\theta, \mathbf{z}) = q(\theta)q(\mathbf{z})$ (Bishop, 2006; Blei & Jordan, 2006). This is known as the mean field approximation for the Bayesian Gaussian Mixture Model. As this assumption imposes restrictions on the form of these distributions, the variational distribution does not necessarily converge to the true posterior. Instead, it converges to the best possible approximation within the constraints of the variational family.

**Prior-Data Fitted Networks.** Müller et al. (2024) introduce a Transformer-based architecture known as Prior-Data Fitted Networks (PFNs), which leverages in-context learning to approximate posterior predictive distributions (PPDs) in supervised learning settings. During training, tasks are sampled from a prior over datasets $p(\mathcal{D})$, producing $\mathcal{D} = \{(x_i, y_i)\}_{i=1}^{N}$, where $x_i \in \mathbb{R}^d$ are feature vectors and $y_i \in \mathbb{R}$ are corresponding targets. These targets can either correspond to classification labels or regression targets. The model is also provided with a query input $x_{\text{test}}$, for which it must predict a distribution over the corresponding output $y$. The PFN is trained to minimize the loss:

$$\ell_\theta = \mathbb{E}_{\mathcal{D} \cup \{x_{\text{test}}, y\} \sim p(\mathcal{D})} \left[ -\log q_\theta(y \mid x_{\text{test}}, \mathcal{D}) \right],$$

where $q_\theta$ denotes the PFN parameterized by $\theta$. In other words, the PFN is trained to predict the target for a test point, given a training set. These are sampled from a dataset. For each batch, new datasets are sampled, and thus the PFN is trained on millions of datasets. Predictions for the test points are made in a single forward pass, meaning the PFN learns to infer each test object's label in one pass. Implicitly, the PFN is performing learning within the forward pass itself, making this a form of meta-learning. Since the PFN is trained on a prior, it can be shown that minimizing this objective is equivalent to minimizing the KL divergence between the model's predictive distribution and the true PPD. Thus, the PFN learns to approximate Bayesian inference and will estimate the posterior distributions.

Crucially, the underlying model that determines $y = f(x)$ can be quite high-dimensional (for example, it can be a multilayer perceptron). The PFN approach avoids estimating these parameters; this is crucial, since focusing on the prediction target, $y$, which is relatively low-dimensional, leads to a less challenging learning problem. Furthermore, the PFN approach models each object $(x, y)$ as a token in the context of the Transformer. For test objects, the $y$ is substituted with zero. Transformer encoder layers then operate on tokens, resulting in a final embedding per object, which is decoded by an MLP to obtain the final prediction for $y$ for the test points.

## 3 CLUSTER-PFN

We first present an overview of the Cluster-PFN and its objectives. Then we discuss its design decisions, and how the Cluster-PFN differs from typical supervised PFNs by Müller et al. (2024).

**Objectives.** The Cluster-PFN should: (1) predict the number of clusters in $[1, K]$ (with a fixed maximum $K$), (2) predict the responsibilities of each object, (3) be able to produce $k \in [1, K]$ clusters when requested by the user, (4) be able to deal with multiple dimensionalities and missing data. First, we discuss how to estimate the number of clusters, which is relatively straightforward. Then we discuss how to estimate the responsibilities, when $k$ is provided or not. This is more challenging; we need to design a PFN that can train even when labels can be exchanged, and we discuss how to estimate responsibilities when the number of clusters is unknown.

**Overview.** The full training pipeline is illustrated in Figure 2. The data $x_1$ to $x_N$ is input to the model, together with $k$, the number of clusters as requested by the user. The output of the Cluster-PFN is a vector of responsibilities per object, indicating the estimated posterior probability of the object belonging to a particular cluster. Inputting $k = 0$ means the Cluster-PFN will predict the posterior over the number of clusters.

**Differences with supervised PFNs.** Unlike supervised learning, unsupervised learning does not provide labels. All feature vectors of all objects are embedded by an MLP. Similar to the PFN, the resulting embeddings represent the objects as tokens in a transformer. The embeddings obtained after the final transformer layer for each object will be used to estimate the responsibilities. Furthermore, there is no training or test set; all objects are processed in the same way. This simplifies the PFN setup. Müller et al. (2024) does not allow attention between test objects. Since we have no train and test split, all objects can attend to each other.

**Estimating the number of clusters.** Estimating the number of clusters is, in fact, a supervised learning task. We generate a synthetic dataset with $k$ clusters, where $k$ is in $[1, 2, \ldots, K]$. Since we know there are $k$ clusters, the dataset has as label $k$. Since the dataset can have a varying number of objects, a transformer is ideal to process it — since it can deal with sequences of varying lengths.

To estimate the number of clusters, we introduce a special object $\rho$ that always has a feature vector of minus ones so the transformer can distinguish it from the normalized input data ($[0, 1]$). We also encode this object using an MLP and concatenate it to the other embeddings to yield an extra token in the context. Since this token should collect information from other tokens, we let each token in the context attend this object, but not the other way around. After the transformer layers, a final MLP decodes this to probabilities for the number of clusters. By a similar argument of Müller et al. (2024), the Cluster-PFN probabilities will approximate the true Bayesian posterior over the number of clusters.

**Estimating responsibilities and dealing with label invariance.** The PFNs of Müller et al. (2024) classify objects into classes. This is done by taking their embeddings at the end of the transformer and using an MLP to turn this into class probabilities. We adopt the same approach, but replace $y$ with the assigned cluster label, which is available to us during synthetic data generation.

However, inherently in clustering, there is permutation invariance between the labels. Since cluster labels in $y$ are arbitrary, the same clustering can be represented by multiple permutations. For training the PFN for classification this is not problematic; the training set provides the matching between the clusters and labels. For clustering, however, we have no train set and test set, and never observe any labels. This ambiguity due to label permutation invariance complicates learning, as the model receives inconsistent supervision across tasks. Because of this, the PFN cannot learn.

To resolve this, we impose a consistent labeling convention during data generation to break the symmetry. Specifically, we select the data point closest to the origin and assign it label 0. We then iteratively assign increasing labels to the clusters based on distance to the origin. This deterministic relabeling enforces a fixed ordering of cluster indices across datasets, so the Cluster-PFN can learn.

**Conditioning on the number of clusters.** The Cluster-PFN should be able to condition on the number of clusters $k$. To this end, $k$ is taken as input to the Cluster-PFN and fed into an MLP. The resulting embedding is added to all objects in the context. This makes sure that all objects are aware of the number of clusters to be used. Note that $k$ is not used to restrict the dimensionality of the posterior of $z$. As such, the Cluster-PFN can, for example, assign objects to 4 clusters, while $k = 3$ is input by the user. When $k = 0$, the PFN will estimate the number of clusters.

**Computing responsibilities when the number of clusters is unknown.** When the user feeds in $k = 0$, the transformer still computes the responsibilities. However, the Cluster-PFN outputs the joint distribution $p(\mathbf{z} \mid \mathbf{X}) = p(\mathbf{z} \mid \mathbf{X}, k = 1)\,p(k = 1) + \ldots + p(\mathbf{z} \mid \mathbf{X}, k = K)\,p(k = K)$ performing full Bayesian inference over a varying number of clusters and their responsibilities. This joint formulation introduces a problem. Since we broke the cluster invariance of the labels, lower cluster labels are observed more often than higher ones. Because of this, later clusters receive disproportionately low responsibilities.

To resolve this, the Cluster-PFN requires two forward passes when $k = 0$ to estimate responsibilities. The first forward pass determines the posterior over the number of clusters. Then, usually, one takes $k$ with maximal posterior probability (user choice). The Cluster-PFN is then run again on the same data, while feeding in the desired $k$. In this case, the correct Bayesian responsibilities are obtained.

**Data generation.** The training process begins with zero-one scaled input data. We provide the true number of clusters via $k$ with probability 0.5, and set $k$ to zero otherwise. This setup allows the model to learn both cluster count prediction as well as conditional clustering to retrieve responsibilities. Everything is a classification task; thus, we use cross entropy as a loss function. Even when $k \neq 0$ the loss with respect to the clusters is always applied. The Cluster-PFN is trained in batches, where each batch consists of multiple clustering datasets. The number of clusters is sampled uniformly in $[1, K]$ for each dataset.

## 4    EXPERIMENTAL SETUP

This section goes over the procedure for the synthetic data generation as well as the baselines and metrics used to evaluate the performance.

**The finite GMM prior.** First, we sample $k \sim \mathcal{U}(1, K)$ and then draw cluster parameters and data points from the Bayesian GMM (see Appendix A for the plate diagram). Let $K$ denote the number of mixture components and $N$ the number of data points. The generative process begins by drawing mixture weights $\boldsymbol{\pi}$ from a Dirichlet prior: $p(\boldsymbol{\pi}) = \mathrm{Dir}(\boldsymbol{\pi} \mid \alpha) = C(\alpha) \prod_{k=1}^{K} \pi_k^{\alpha-1}$, where $C(\alpha_0)$ is a normalization constant that ensures the mixing proportions sum up to 1. This essentially provides the prior probability of sampling from each cluster. For each component $i \in \{1, 2, \ldots, k\}$, a mean $\boldsymbol{\mu}_i$ and covariance $\boldsymbol{\Sigma}_i$ are sampled from a Normal-Inverse-Wishart prior. This is a standard conjugate prior for GMMs, which makes the VI approximation much simpler. Note that this is not a necessary condition for the PFN and favors VI.

Finally, for each data point $i \in \{1, \ldots, N\}$, a latent assignment $z_i \sim \mathrm{Categorical}(\boldsymbol{\pi})$ is drawn, indicating the cluster membership, and the observation $x_i \sim \mathcal{N}(\boldsymbol{\mu}_{z_i}, \boldsymbol{\Sigma}_{z_i})$ is sampled from the corresponding Gaussian component. To extend Cluster-PFN to handle missingness, we apply random masking during training: in each dataset, a random proportion of entries (uniformly sampled between 0–80%) is set to missing. For more details about the distributions and hyperparameters, refer to Appendix B

**Trained models.** We train Cluster-PFN models under three input settings: (i) 2D inputs, (ii) inputs ranging from 2D to 5D (lower-dimensional inputs are zero-padded to a consistent 5D size), and (iii) 5D inputs with missing values. For each setting, we train two models: one on Easy clusters and one on Hard clusters. Cluster difficulty is determined by the $\beta$ hyperparameter: $\beta = 0.01$ yields well-separated (Easy) clusters, while $\beta = 0.1$ produces overlapping (Hard) clusters. All other hyperparameters are fixed at $\alpha = 0.1$, $m = 0$, $W = I$, and $v = d$, where $d$ is the input dimension.

Each epoch samples 100 datasets, with dataset sizes drawn as $N \sim \mathcal{U}(100, 500)$. 2D models are trained for 600,000 epochs over 40 GPU hours, while 2D-5D models are trained for 900,000 epochs over 60 GPU hours. All experiments are conducted on an NVIDIA L40 GPU. Additional architectural details are provided in Appendix C.

**Baselines for predicting the number of clusters** To evaluate the performance of the Cluster-PFN in predicting the number of clusters, we compare it against the GMM using three standard model selection criteria: the Akaike Information Criterion (AIC) (Akaike, 1974), the Bayesian Information Criterion (BIC) (Schwarz, 1978), and the Silhouette Score (Rousseeuw, 1987). For each of these handcrafted criteria, we fit a GMM across a range of candidate cluster numbers from 2 to $K$, excluding 1 since the silhouette score is undefined for a single cluster, and select the cluster count that yields the optimal score. This also implies that, in our dataset evaluations, no dataset will contain only one cluster."

For VI, we follow the approach of Bishop (2006). Specifically, we fit models with components ranging from 2 to $K$ and select the one with the highest variational lower bound. To account for the multimodality of the posterior, we include an additional penalty term of $\ln(k!)$ for each candidate $k$.

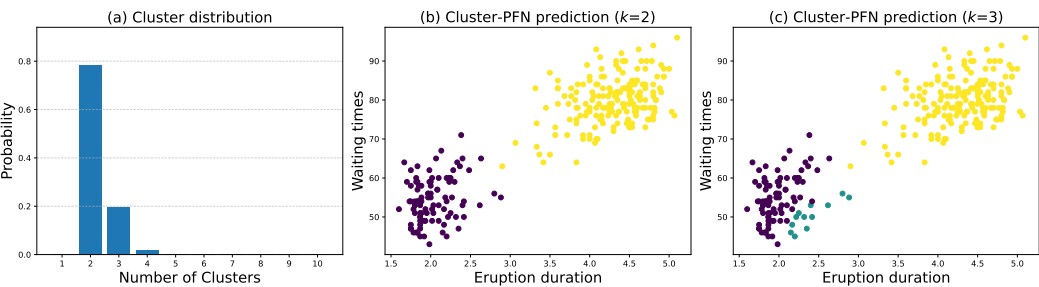

Figure 3: Cluster-PFN prediction on the Old Faithful dataset (Cluster-PFN trained on Easy prior).

**Evaluating clustering performance.**   Since both the Cluster-PFN and VI produce probabilistic cluster assignments, we evaluate model performance using metrics that capture hard and soft label clustering. To obtain the hard cluster labels, we first perform model selection to determine the number of clusters. We then fit the model and assign each data point to the cluster with the highest posterior probability. When just comparing the Cluster-PFN and VI, we also include $k = 1$ cluster.

For hard cluster assignments, we employ three standard external metrics: Adjusted Rand Index (ARI) (Hubert & Arabie, 1985a), Adjusted Mutual Information (AMI) (Vinh et al., 2010), and Purity (Manning et al., 2008). ARI quantifies the agreement between predicted and true labels, adjusted for chance. AMI measures the mutual dependence between predicted and true labels, also adjusted for chance. Purity assesses the extent to which predicted clusters contain points from a single ground-truth class. To evaluate the probabilistic predictions, we compute the Negative Log-Likelihood (NLL) of the true cluster assignments under the model's predicted posterior distribution. Since the cluster labels produced by the models may be permuted, we compute the NLL under all possible label permutations and report the minimum value obtained. Additional details of these metrics can be found in Appendix D.

**Real world datasets and studying robustness to missingness.**   To evaluate clustering under missingness, we use four real-world genomic benchmark datasets: one RNA sequencing dataset (Network et al., 2013) and three GWAS summary statistics datasets (Julienne et al., 2020), assessing external metric scores. These datasets were also used in prior work by McCaw et al. (2022) to study missingness, which motivated our choice, though that study applied a standard GMM rather than a Bayesian approach. For each dataset of size $N \times d$, we introduce missingness by masking a proportion of random elements. Each higher missingness level builds on the previous mask, while ensuring at least one element per row remains observed.

Among the Cluster-PFN models trained on the Easy and Hard priors, the one trained on the Hard prior performed best on real-world benchmarks. We evaluated it against baseline methods, applying feature-wise mean or median imputation on missing values (reporting only median results, as both gave similar outcomes) and using handcrafted model selection to assign cluster labels. VI was provided with the same priors used for the hard datasets.

## 5   RESULTS

**RQ1.1: Clustering real-world data.**   We begin by qualitatively analyzing Cluster-PFN's clustering on the Old Faithful dataset, a classic real-world dataset. It records eruption durations and waiting times (Corduneanu & Bishop, 2001), and we use it to visually assess the Cluster-PFN's performance. Figure 3 shows the clustering results produced by Cluster-PFN trained on the Easy prior on the Old Faithful dataset. The model assigns the highest posterior probability to 2 clusters, followed by 3 clusters. The partitionings for the clustering conditioned on 2 and 3 clusters are also shown, demonstrating that both align well with the underlying structure of the data.

**RQ1.2: Clustering with user-supplied condition on synthetic data.**   We also qualitatively evaluate the model's behavior when conditioned on a specified number of clusters. Figure 4 illustrates the Cluster-PFN's predictions on data sampled from the prior (4a) under different conditioning scenar-

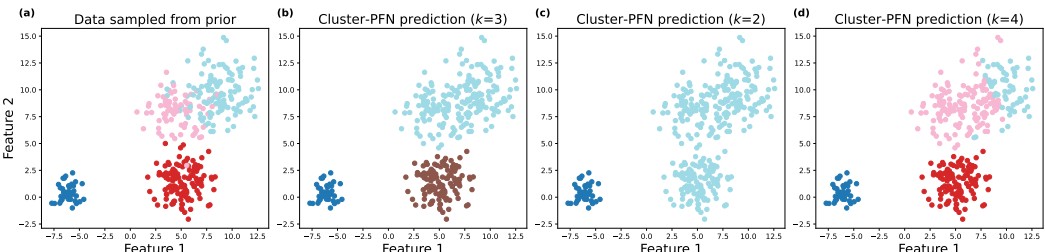

Figure 4: Cluster-PFN predictions on prior-data for different conditions (Easy prior).

Table 1: % of correct cluster counts predicted across 1000 datasets (E = Easy, H = Hard).

Table 2: Cluster count accuracy and runtime for 1000 2D Easy datasets.

| Model | 2D E | 5D E | 2D H | 5D H | Model | Accuracy (% ↑) | Time (s↓) |
|---|---|---|---|---|---|---|---|
| Cluster-PFN | **64** | **72** | **44** | **52** | Cluster-PFN | **64** | **0.02** |
| GMM (AIC) | 36 | 41 | 29 | 31 | VI (1 init.) | 42 | 1.06 |
| GMM (BIC) | 42 | 41 | 29 | 28 | VI (10 inits.) | 51 | 10.39 |
| GMM (SIL) | 21 | 25 | 12 | 14 | VI (50 inits.) | 54 | 52.19 |
| VI (1 init) | 42 | 30 | 32 | 26 | VI (100 inits.) | 54 | 103.66 |

ios. The model assigned the highest posterior probability to three clusters (0.92), failing to capture the true distribution of four clusters. It assigned lower probabilities to four clusters (0.06) and two clusters (0.002). When conditioned on these cluster counts, the figure illustrates the model's ability to generate coherent and flexible clusterings. However, Cluster-PFN does not always adhere strictly to the conditioning; its adherence is further analyzed in Appendix E.1. Especially, if it is clear there are $k$ clusters, but it is conditioned on a wildly different number, it may fail to adhere to the conditioning.

**RQ2: Comparison to GMM and VI for estimating the number of clusters** To evaluate the accuracy of the Cluster-PFN's cluster count predictions, we compare its performance against established handcrafted methods and VI. We sample 1000 datasets and measure the proportion of instances in which each method correctly predicts the true number of clusters. Table 1 reports the percentage of correctly predicted cluster counts across different models. Cluster-PFN consistently achieves the highest accuracy, outperforming all baseline methods in every setting. Overall, these findings underscore Cluster-PFN's strong capability to recover the true number of clusters across varying levels of problem difficulty and dimensionality.

This experiment was initially conducted with VI using a single initialization. We examine how increasing the number of initializations impacts cluster prediction performance, analyzing the trade-off between runtime and accuracy. For this analysis,we sample 1,000 datasets from the 2D Easy setting and compare Cluster-PFN against VI with varying initialization counts. All VI runs are executed sequentially. Table 2 displays Cluster-PFN obtaining the highest accuracy and being 50 times faster than the fastest VI configuration. Increasing the number of VI initializations improves accuracy marginally and results in a linear growth in runtime.

**RQ3: Clustering quality and uncertainty estimation of Cluster-PFN and VI** To evaluate the cluster assignment performance of Cluster-PFN, we sample 1,000 datasets and compute the ARI, AMI, Purity, and NLL scores for both Cluster-PFN and VI (with 10 initializations).

As shown in Figure 5a, the violin plot for the 2D Hard dataset illustrates that the distribution of external metric scores (AMI, ARI, Purity) is similar between Cluster-PFN and VI. The corresponding NLL histogram in Figure 5c shows a similar pattern, with most values concentrated below 1. Unlike VI, Cluster-PFN assigns a nonzero probability to every cluster, so its NLL values never reach exactly 0 or infinity. This explains the last bar in the VI histogram, which contains a considerable number of infinite values — these occur when VI underestimates the number of clusters. The experiments so far were conducted on datasets containing 100 to 500 points.

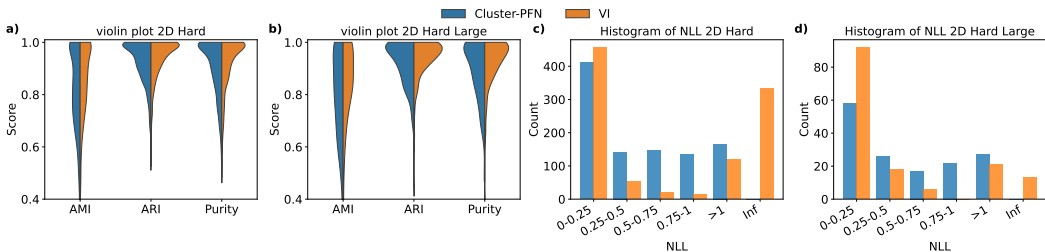

Figure 5: External metrics and NLL on 2D Hard datasets with model selection: (a)–(b) show metric distributions for small and large datasets, (c)–(d) show corresponding NLL histograms.

To evaluate Cluster-PFN's ability to generalize to larger datasets, we tested 150 datasets, each containing 10,000 data points (5b and 5d). Both plots show that Cluster-PFN maintains performance comparable to VI, with similar distributions across external metrics and NLL values. This demonstrates that Cluster-PFN, despite being trained on smaller datasets, can scale effectively while preserving accurate clustering quality. We also evaluated the inference speed of Cluster-PFN and VI on the 5D Hard datasets across varying dataset sizes, using 10 initializations for VI and 100 sampled datasets. Cluster-PFN was consistently faster: for 100 points, it took approximately 0.03 s versus 4 s for VI; for 5,000 points, approximately 4 s versus 134 s; and for 20,000 points, approximately 60 s versus 410 s.

We do not have space for all experiments here. In this study, model selection was applied to Cluster-PFN and VI to determine the cluster count. Appendix E.2 shows violin plots and histograms for the remaining Easy and Hard priors, both when the number of clusters is unknown and when provided, confirming the results in Figure 5. Appendix E.3 reports the percentage of datasets where one model outperforms the other for each metric; VI often achieves more wins, but ties are frequent. We also compared Cluster-PFN with GMM, K-means++ by ranking models across external metrics on sampled datasets; Cluster-PFN consistently outperforms these baselines, as shown in Appendix E.4.

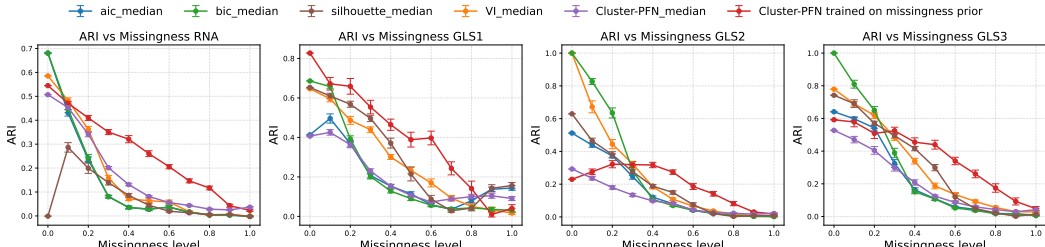

Figure 6: ARI scores of different models at varying levels of missing data in real-world datasets. {model}_median indicates the models were fitted on the median-imputed dataset. Error bars represent the standard error across 20 simulations (higher scores indicate better clustering).

**RQ4: Performance of missingness on real-world data.**    Figure 6 shows ARI versus missingness. At low missingness, both standard and missingness-trained Cluster-PFN underperform the baselines on 3/4 datasets, but on GLS1, the missingness-trained model outperforms nearly all methods across most levels. For 20–30% missingness and above, the missingness-trained Cluster-PFN outperforms all baselines and declines more slowly. Notably, the standard Cluster-PFN performs poorly even at 0% missingness, worse than the missingness-trained model, suggesting that the Cluster-PFN learns more effectively under more challenging conditions offered by the missingness prior. See Appendix E.5 for AMI, Purity, and results on mean-imputed datasets, which show similar patterns.

## 6 DISCUSSION

The results are clear: when predicting the number of clusters, the performance of the Cluster-PFN shines and is significantly better than previous approaches, such as VI, BIC, AIC, and the Silhouette

score. On data from the prior, the Cluster-PFN mostly matches the performance of VI. However, for real-world data, the Cluster-PFN is not always competitive, only offering clear benefits on the GLS1 dataset. The Cluster-PFN however does perform well for high missingness, illustrating the power of complex priors that integrate missingness which can be effortlessly integrated into the Cluster-PFN. Suboptimal performance observed on GLS2 and GLS3 may arise from prior-misspecification: since the PFN is trained only on data from the GMM prior, it may suffer a domain shift when encountering real-world data that does not adhere well to the prior. This was also observed recently by Viering et al. (2025). Solutions are developing more complex priors to increase the training data diversity; or real-world data can also be integrated into PFN training.

We have argued before that the Cluster-PFN approximates the true Bayesian posterior over the number of clusters — without relying on any further assumptions. Responsibilities are estimated independently, making them conditionally independent—a weaker approximation than VI, which factorizes over $\theta$ and $z$. In principle, a transformer could avoid approximations to $z$ by capturing the full joint posterior in an auto-regressive manner. This would involve sequentially decoding posterior predictions, each conditioned on previously assigned points. The major downside of this approach is that it would require one forward pass per object. Meanwhile, the Cluster-PFN of our design only requires one or two forward passes, making it computationally attractive. Finally, the Cluster-PFN does not provide the outputs $\theta$. This is a conscious choice; $\theta$ is high-dimensional and has strong dependencies (thus requiring auto-regressive output), and therefore we hypothesize it will be harder to learn. By focusing on responsibilities, we use the same trick as PFNs: low-dimensional outputs for more easy learning. An auto-regressive approach, on the other hand, would not require any symmetry breaking that we require, and may offer improved approximation.

One clear downside to our framework and that of the PFN in general is that it requires a large up-front cost to train the transformer for a particular prior. If one is interested in changing priors during the analysis, VI and other approximations are more natural. One line of work is Whittle et al. (2025), who develop PFN-like models for multiple priors, which can be integrated into our work as well.

Finally, PFN models are often compared to MCMC in terms of speed. To our knowledge, this is the first time that PFNs are compared to the VI-approximation, which is especially practically relevant since VI is typically much faster than MCMC. Even in this case, we find that PFNs can offer further speed advantages while closely matching VI's performance. It should be noted that the conjugate prior chosen in our experiments favors VI, since it makes the VI approximation tractable and fast.

## 7    SUMMARY, LIMITATIONS AND FUTURE RESEARCH

We presented Cluster-PFN, a Transformer-based model for computing posterior cluster assignments. Cluster-PFN predicts cluster membership probabilities, estimates the number of clusters, and supports conditional clustering when a cluster count is specified. Our experiments demonstrate that it produces accurate and coherent clusterings, adapts effectively to conditioning, and consistently outperforms traditional handcrafted methods (AIC, BIC, silhouette) and VI in estimating the number of clusters. Additionally, it achieves competitive performance with VI while offering faster inference and strong generalization to larger datasets. When missingness reaches 30% or more, Cluster-PFN outperforms all baseline methods for the real-world datasets.

Currently, Cluster-PFN is invariant to sample permutations but not to feature permutations. Incorporating feature invariance, as explored in Arbel et al. (2025); Hollmann et al. (2025), could improve robustness and generalization. For these architectures, the PFN can also be applied to dimensions higher than encountered during training. Additionally, scaling Cluster-PFN to handle higher-dimensional inputs would be a valuable direction for assessing its performance on more complex datasets. Especially in bio-informatics, it seems there are popular priors for count data (negative-binomial mixture models, see (Wade, 2023)) — it would be useful to train Cluster-PFNs specialized for such applications. We have stuck to a fairly standard GMM prior that is conjugate so that VI simplifies and that is well-known; our Cluster-PFN opens the door to investigations of highly complex priors that could not be studied before due to computational concerns. Finally, our prior sticks to a finite GMM prior with a bounded number of clusters, which allows us to approach predicting the cluster count as a classification problem. How to extend our work to mixture models of unbounded size (e.g. Dirichlet process) remains a challenging open question.

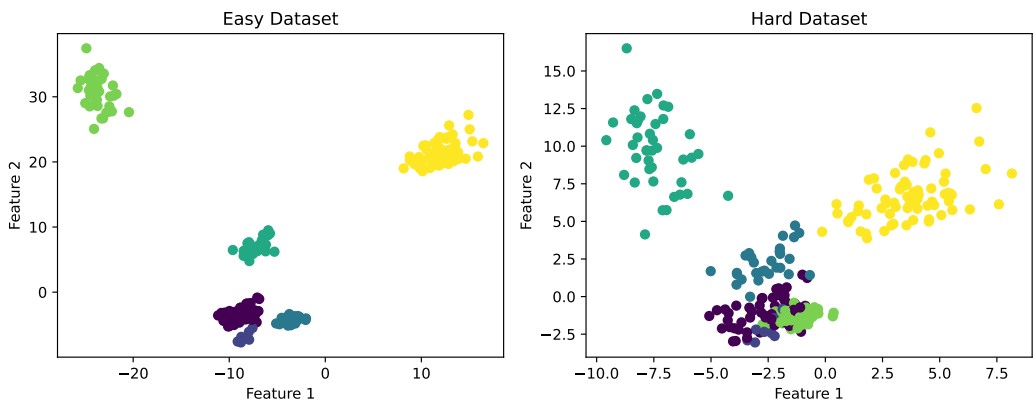

Figure 7: Example of datasets sampled when $\beta = 0.01$ (Easy) and $\beta = 0.1$ (Hard)

## 8 REPRODUCIBILITY STATEMENT

To ensure reproducibility, we fixed random seeds throughout all stages of our experiments. This includes synthetic data generation, evaluation of the models, as well as the creation of missingness masks, ensuring that results can be consistently replicated.

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

## A   PRIOR DATA GENERATION

This section provides the Bayesian plate diagram of the prior.

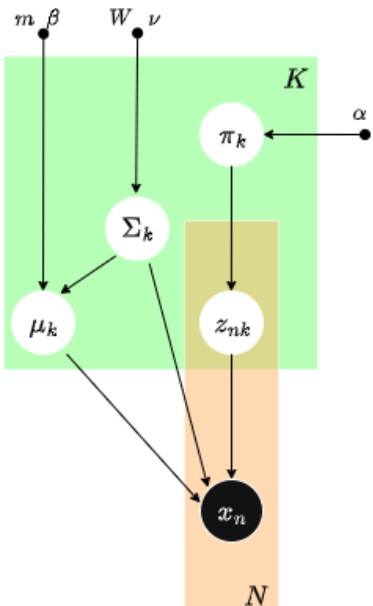

## B   DISTRIBUTIONS

Here, we provide more details on the Dirichlet Distribution and Wishart distribution.

### B.1   DIRICHLET DISTRIBUTION

The Dirichlet is a multivariate distribution, over $K$ random variables $0 \leq \mu_k \leq 1$, $\sum_{k=1}^{K} = 1$. Denoting $\mu = (\mu_1, ..., \mu_k)^T$ and $\alpha = (\alpha_1, .., \alpha_n)^T$, we have

$$p(\pi) = \text{Dir}(\pi | \alpha_0) = C(\alpha_0) \prod_{k=1}^{K} \pi_k^{\alpha_0 - 1} \tag{2}$$

where $C(\alpha) = \frac{\Gamma(\hat{\alpha})}{\Gamma(\hat{\alpha}_1)...\Gamma(\hat{\alpha}_K)}$ and $\Gamma(\alpha) = \int t^{\alpha-1} e^{-t} dt$

The purpose of $C(\alpha)$ is that it serves as a normalization constant so the pdf integrates to 1.

### B.2   WISHART DISTRIBUTION

$$\mathcal{W}(\Lambda \mid W, \nu) = B(W, \nu) |\Lambda|^{(\nu - D - 1)/2} \exp\left(-\frac{1}{2} \text{Tr}(W^{-1}\Lambda)\right) \tag{3}$$

Where

$$B(W, \nu) = |W|^{-\nu/2} \left(2^{\nu D/2} \pi^{D(D-1)/4} \prod_{i=1}^{D} \Gamma\left(\frac{\nu + 1 - i}{2}\right)\right)^{-1} \tag{4}$$

Where $W$ is a $D \times D$ symmetric, positive definite matrix. The parameter $\nu$ is called the number of degrees of freedom and is restricted to $v \geq D$.

## C  FURTHER EXPERIMENTAL SETUP AND TRAINING DETAILS

We provide all the details regarding the architecture, and we also show some of the loss curves that we observe during training.

### C.1  ARCHITECTURE

The Cluster-PFN architecture is fully based on the PyTorch Paszke et al. (2019) library and utilizes the encoder component of the Transformer. It employs an embedding dimension of 256 and a hidden dimension of 512. The model uses 4 attention heads and consists of 4 encoder layers. Each encoder layer includes the following components:

1. **Multi-Head Self-Attention** with 4 attention heads
2. **Add & Layer Normalization** applied after the attention mechanism
3. **Position-wise Feedforward Network**, consisting of:
   - A linear layer: $256 \rightarrow 512$
   - ReLU activation
   - A linear layer: $512 \rightarrow 256$
4. **Add & Layer Normalization** applied after the feedforward network

After passing through the encoder, a final layer maps each input to 10 output values. Each output represents the probability of the data point belonging to a specific cluster.

We use a cosine annealing (Loshchilov & Hutter, 2016) learning rate schedule with a warm-up, applied to an AdamW optimizer initialized with a tuned learning rate of 0.001.

### C.2  LOSS FUNCTION

The loss function used is the cross-entropy loss. There are two components to the overall loss: the loss from the data point cluster assignments and the loss from the predicted number of clusters. The final loss is computed as the sum of the average cluster assignment loss and the cluster prediction loss.

## D  EXTERNAL METRIC FORMULAS

Here, we discuss the meaning of the metrics in more detail and their interpretation.

### D.1  ADJUSTED RAND INDEX

$$\text{RI} = \frac{TP + TN}{TP + FP + FN + TN} \tag{5}$$

The Rand Index measures the similarity between clusterings by examining pairs of data points:

- **TP (True Positives)**: Pairs of points that are in the same cluster in both the predicted and true clusterings.
- **TN (True Negatives)**: Pairs of points that are in different clusters in both the predicted and true clusterings.
- **FP (False Positives)**: Pairs of points that are in the same cluster in the prediction, but in different clusters in the ground truth.
- **FN (False Negatives)**: Pairs of points that are in different clusters in the prediction, but in the same cluster in the ground truth.

Intuitively, the Rand Index evaluates the proportion of decisions (about whether a pair of points should be in the same cluster or not) that the clustering algorithm got correct. However, since the RI does not account for chance groupings, the Adjusted Rand Index introduces a normalization that adjusts the score for random labelings (Hubert & Arabie, 1985b).

## D.2 ADJUSTED MUTUAL INFORMATION

$$\text{MI}(U, V) = \sum_{i=1}^{|U|} \sum_{j=1}^{|V|} \frac{|U_i \cap V_j|}{N} \log \left( \frac{N|U_i \cap V_j|}{|U_i||V_j|} \right) \tag{6}$$

Where:

- $U = \{U_1, U_2, \ldots, U_{|U|}\}$ is the set of ground truth clusters.
- $V = \{V_1, V_2, \ldots, V_{|V|}\}$ is the set of predicted clusters.
- $N$ is the total number of data points.
- $|U_i \cap V_j|$ is the number of data points that appear in both cluster $U_i$ and cluster $V_j$.
- $|U_i|$ and $|V_j|$ are the sizes of clusters $U_i$ and $V_j$, respectively.

The higher the mutual information, the better the clustering. However, mutual information is not normalized and can be biased by the number of clusters. Adjusted mutual information corrects this bias by subtracting the expected MI of random clusterings and normalizing the result Pedregosa et al. (2011).

## D.3 PURITY

$$\text{Purity} = \frac{1}{N} \sum_{k=1}^{K} \max_{j} |C_k \cap L_j| \tag{7}$$

where $N$ is the total number of data points and $K$ is the number of clusters. $C_k$ is the set of data points in cluster $k$, and $L_j$ as the set of data points with ground truth label $j$. The term $|C_k \cap L_j|$ represents the number of data points in cluster $k$ that belong to ground truth class $j$. In essence, purity measures the extent to which clusters contain data points predominantly from a single true class. A high purity value indicates that most points within each cluster belong to the same class, reflecting better clustering performance.

## D.4 NEGATIVE LOG-LIKELIHOOD

$$l_{NLL} = - \sum_{i=1}^{N} \sum_{k=1}^{K} y_{ik} \log(p_{ik}) \tag{8}$$

We sum over all data points and clusters. For each input, $y_{ik}$ is a binary indicator that is 1 if the sample, $i$ belongs to the cluster $k$ and zero otherwise, and $p_{ik}$ is the predicted probability that sample $i$ belongs to cluster $k$. Since dataset sizes can vary, we use the average NLL as our loss metric. As our true labels will not necessarily match with the labels from the models, we will permute the labels and retrieve the lowest NLL given by both of the models.

## E FURTHER EXPERIMENTS

### E.1 ADHERING TO THE CONDITIONING MECHANISM

To quantitatively evaluate the conditioning accuracy, we sampled 30,000 synthetic datasets and provided a randomly selected cluster count as the conditioning input to the Cluster-PFN. We then measured how closely the predicted number of clusters matched the specified condition. This was done by first obtaining the hard labels from the Cluster-PFN's predictions and then counting the number of unique labels. The evaluation was performed under varying thresholds:

- A threshold of 0 means the model predicted exactly the conditioned number of clusters.
- A threshold of 1 allows for a deviation of ±1 from the condition.
- A threshold of 2 allows for a deviation of ±2 from the condition.

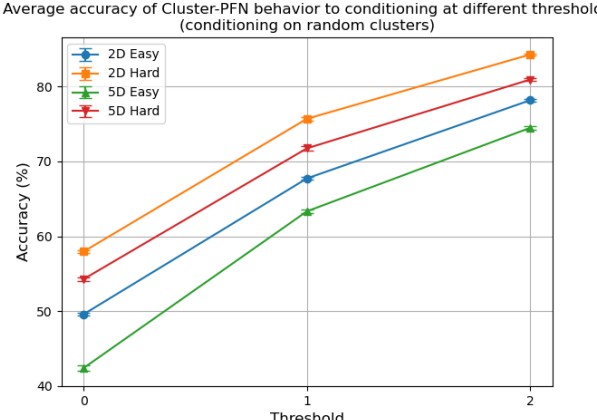

Figure 8: Cluster-PFN listening to condition at different thresholds (conditioning on random clusters)

Figure 8 shows the accuracy across different thresholds. In the Hard setting, the Cluster-PFN adheres to the conditioning slightly better than in the Easy setting. However, the conditioning is generally ineffective when using randomly specified cluster numbers, as evidenced by the highest accuracy at threshold 0 being just under 60%.

However, randomly sampling condition cluster values may not be the most meaningful way to evaluate the model's conditioning ability, as some randomly assigned conditions may differ substantially from the Cluster-PFN's natural prediction. In such cases, forcing the model to deviate significantly from its prediction is not very sensible.

To address this, we conduct a second experiment. Instead of randomly selecting conditions, we begin with the model's unconditioned prediction and perturb it by ±1 cluster. We then evaluate how well the Cluster-PFN adapts to these nearby, more reasonable conditioning values.

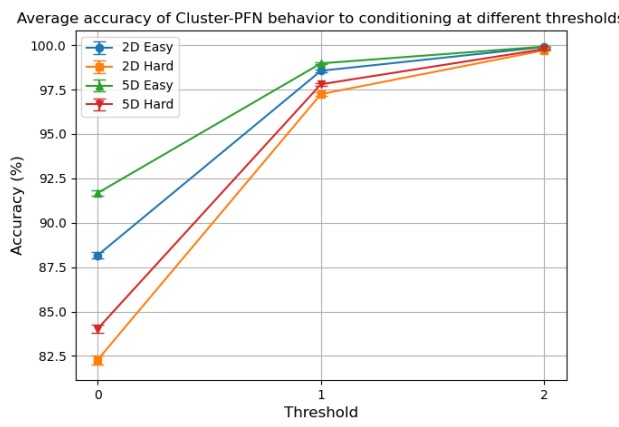

Figure 9: Cluster-PFN listening to condition at different thresholds (conditioning on more appropriate clusters)

We observe significantly higher accuracy when the conditioning values are more reasonable. The model already achieves strong accuracy at a threshold of 0, with a sharp increase in performance at a threshold of 1, and reaches perfect accuracy at the final threshold.

## E.2   EXTERNAL METRIC VIOLIN AND HISTOGRAM PLOT OF CLUSTER-PFN AGAINST VI

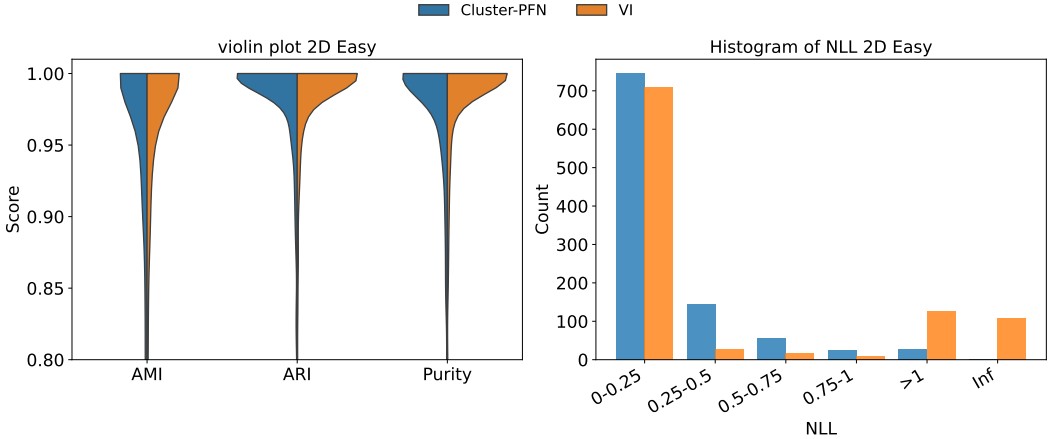

Figure 10: External metrics and NLL for 2D Easy datasets when the number of clusters is unknown.

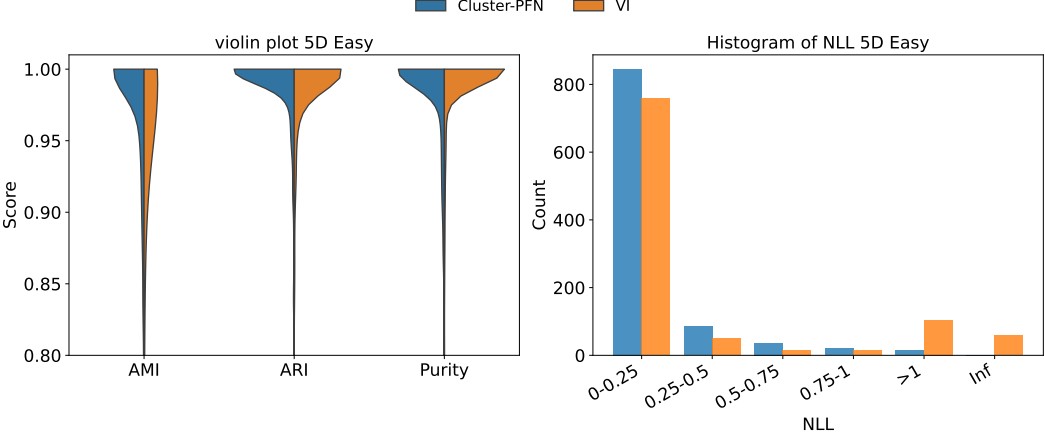

Figure 11: External metrics and NLL for 5D Easy datasets when the number of clusters is unknown.

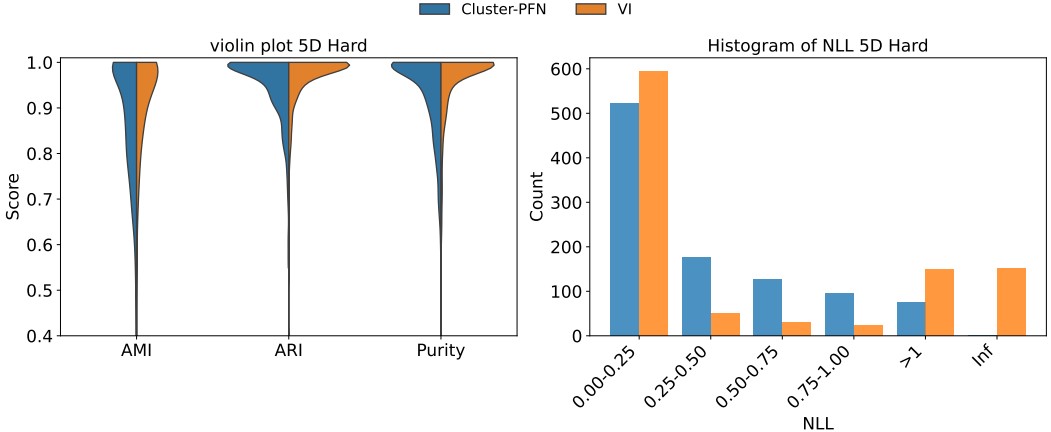

Figure 12: External metrics and NLL for 5D Hard datasets when the number of clusters is unknown.

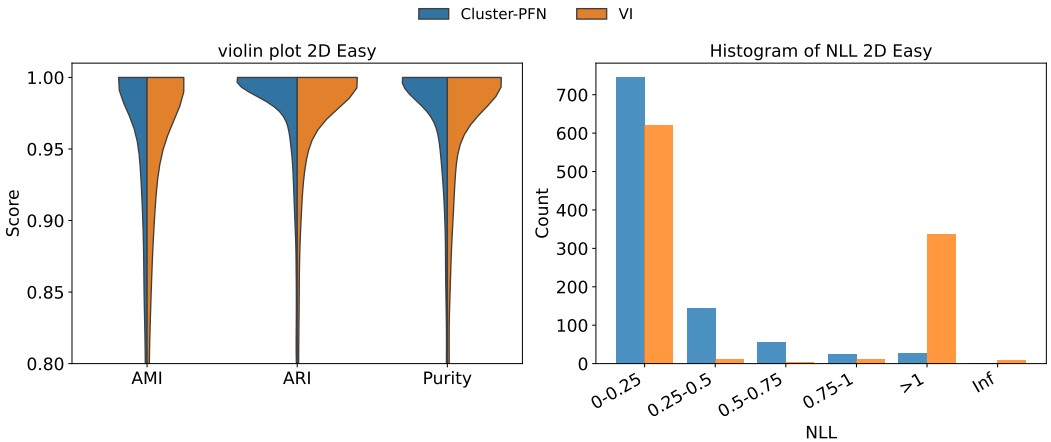

Figure 13: External metrics and NLL for 2D Easy datasets when the number of clusters is known.

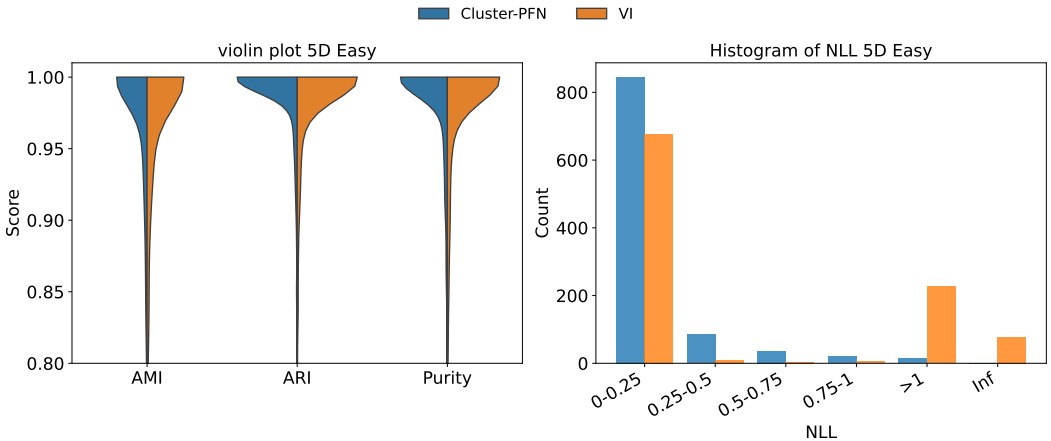

Figure 14: External metrics and NLL for 5D Easy datasets when the number of clusters is known.

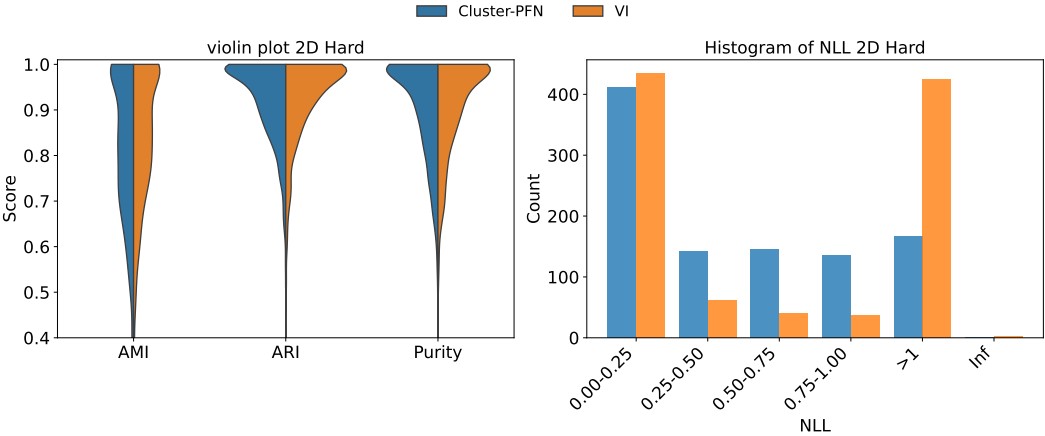

Figure 15: External metrics and NLL for 2D Hard datasets when the number of clusters is known.

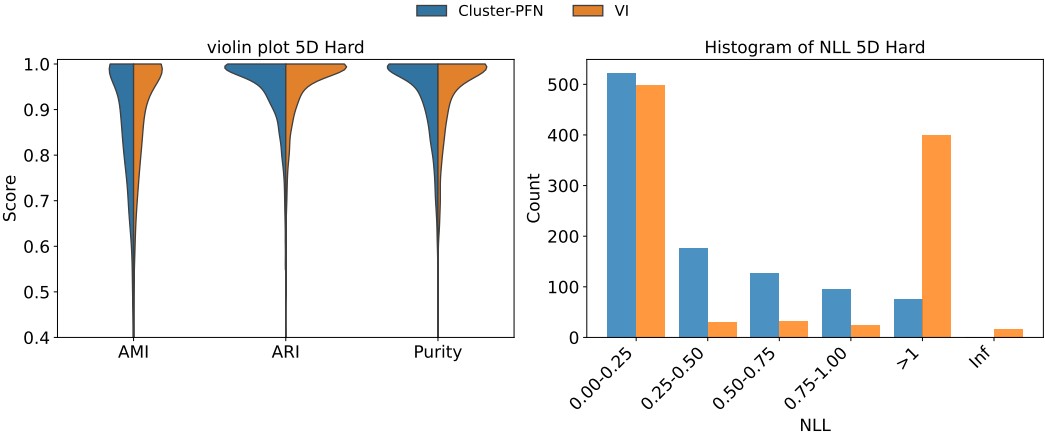

Figure 16: External metrics and NLL for 5D Hard datasets when the number of clusters is known.

### E.3 EXTERNAL METRIC WIN RATE SCORES OF CLUSTER-PFN AGAINST VI

Table 3: Win rate across 1,000 datasets with both models given the true number of clusters.

| Setting | Model | Experiment A: VI 1 init | | | | Experiment B: VI 10 init | | | |
|---|---|---|---|---|---|---|---|---|---|
| | | AMI | ARI | Purity | NLL | AMI | ARI | Purity | NLL |
| 2D Easy | Cluster-PFN | 39.1 | 40.0 | 37.2 | 59.5 | 25.8 | 25.9 | 23.6 | 38.6 |
| | VI | 22.8 | 20.4 | 18.2 | 49.5 | 32.3 | 29.4 | 26.4 | 61.4 |
| | Ties | 38.1 | 39.6 | 44.6 | 0 | 41.9 | 44.7 | 50 | 0 |
| 5D Easy | Cluster-PFN | 38.7 | 39.6 | 35.4 | 48.0 | 24.3 | 24.2 | 20.7 | 33.3 |
| | VI | 18.6 | 15.2 | 14.0 | 52.0 | 26.5 | 22.7 | 21.4 | 66.7 |
| | Ties | 42.7 | 45.2 | 50.6 | 0 | 49.2 | 53.1 | 53.1 | 0 |
| 2D Hard | Cluster-PFN | 36.4 | 41.4 | 39.4 | 59.1 | 23.8 | 29.3 | 28.1 | 49.9 |
| | VI | 40.9 | 34.7 | 32.9 | 40.9 | 52.4 | 45.4 | 42.9 | 51.1 |
| | Ties | 22.7 | 23.9 | 27.7 | 0 | 23.8 | 25.3 | 29 | 0 |
| 5D Hard | Cluster-PFN | 34.8 | 39.6 | 37.1 | 58.3 | 20.3 | 24.6 | 22.4 | 47.3 |
| | VI | 39.6 | 33.6 | 32.0 | 41.7 | 51.8 | 46.3 | 44.1 | 52.7 |
| | Ties | 25.6 | 26.8 | 30.9 | 0 | 27.9 | 29.1 | 33.5 | 0 |

Table 4: Win rate across 1,000 datasets with both models not given the true number of clusters

| Setting | Model | Experiment A: VI 1 init | | | | Experiment B: VI 10 init | | | |
|---|---|---|---|---|---|---|---|---|---|
| | | AMI | ARI | Purity | NLL | AMI | ARI | Purity | NLL |
| 2D Easy | Cluster-PFN | 27.2 | 28.10 | 15.5 | 40 | 18.5 | 17.6 | 8.3 | 30.8 |
| | VI | 30.3 | 26.8 | 29.4 | 60 | 36.1 | 33.3 | 34.5 | 69.2 |
| | Ties | 42.5 | 45.1 | 55.1 | 0 | 45.4 | 49.1 | 57.2 | 0 |
| 5D Easy | Cluster-PFN | 41.9 | 42.4 | 11 | 53.6 | 34.9 | 34.5 | 5.9 | 42.2 |
| | VI | 25 | 22.5 | 25 | 46.4 | 29.1 | 26.3 | 28.2 | 57.8 |
| | Ties | 33.1 | 35.1 | 64 | 0 | 36 | 39.2 | 65.9 | 0 |
| 2D Hard | Cluster-PFN | 21.4 | 25.9 | 22.9 | 53.8 | 14.9 | 19.2 | 18.4 | 49.9 |
| | VI | 53.1 | 47.2 | 44.7 | 46.2 | 59.7 | 53.8 | 49.9 | 50.1 |
| | Ties | 25.5 | 26.9 | 32.4 | 0 | 25.4 | 27 | 31.7 | 0 |
| 5D Hard | Cluster-PFN | 26.1 | 30.2 | 15 | 54.2 | 18.2 | 21.6 | 9.9 | 45.1 |
| | VI | 51.7 | 46.2 | 49.1 | 45.8 | 58.6 | 54.0 | 54.7 | 54.9 |
| | Ties | 22.2 | 23.6 | 35.9 | 0 | 23.2 | 24.4 | 35.4 | 0 |

Tables 3 and 4 report the percentage of datasets where each model outperforms the other, both when the true number of clusters is provided and when it must be inferred through model selection. We evaluate VI with both a single initialization and 10 initializations for a fairer comparison.

Table 3 shows that when the true number of clusters is known and VI is run with only one initialization, Cluster-PFN often outperforms VI—either achieving higher win rates directly or tying more frequently. With 10 initializations, however, VI gains a clear advantage, particularly in the harder settings, though ties remain common across many metrics.

Table 4 highlights the more challenging scenario where the number of clusters is not provided. Here, VI generally outperforms Cluster-PFN under both initialization settings. Nevertheless, a large proportion of results still end in ties, indicating that Cluster-PFN remains competitive.

### E.4 EXTERNAL METRIC SCORES OF CLUSTER-PFN AGAINST GMM AND K-MEANS++

We also evaluate the Cluster-PFN against traditional clustering algorithms such as GMM and K-means++. Using 30,000 sampled datasets, we evaluate each model by computing its average ranks across the external metrics (AMI, ARI, purity), where a rank of 1 denotes the best performance and 3 the worst. The GMM and K-means++ were provided with the true number of clusters during evaluation.

Table 5: Mean AMI ranks of the models across 30,000 datasets for various experiments (lower is better).

| Model | Rank (↓) | | | |
|---|---|---|---|---|
| | 2D Easy | 5D Easy | 2D Hard | 5D Hard |
| Cluster-PFN | **1.57** | **1.57** | **1.60** | **1.56** |
| GMM | 1.96 | 2.02 | 1.75 | 1.72 |
| K-means++ | 2.47 | 2.41 | 2.65 | 2.72 |

| Model | 2D Easy | 5D Easy | 2D Hard | 5D Hard |
|---|---|---|---|---|
| Cluster-PFN | **1.54** | **1.54** | **1.50** | **1.49** |
| GMM | 1.99 | 2.05 | 1.82 | 1.79 |
| K-means++ | 2.47 | 2.41 | 2.68 | 2.72 |

Table 6: Mean ARI ranks of the models across 30,000 datasets for various experiments (lower is better).

| Model | 2D Easy | 5D Easy | 2D Hard | 5D Hard |
|---|---|---|---|---|
| Cluster-PFN | **1.71** | **1.70** | **1.70** | **1.66** |
| GMM | 1.93 | 1.99 | 1.72 | 1.70 |
| K-means++ | 2.36 | 2.31 | 2.58 | 2.64 |

Table 7: Mean Purity ranks of the models across 30,000 datasets for various experiments (lower is better).

### E.5 ADDITIONAL MISSINGNESS AND REAL-WORLD RESULTS

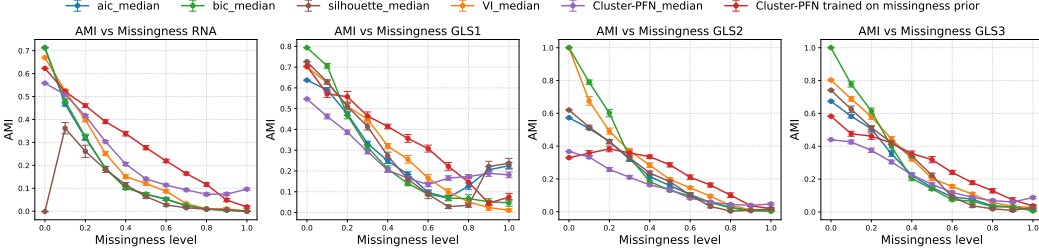

Figure 17: AMI scores for different models at varying missingness levels in real-world median imputed datasets. Error bars show standard error across 20 simulations (higher is better).

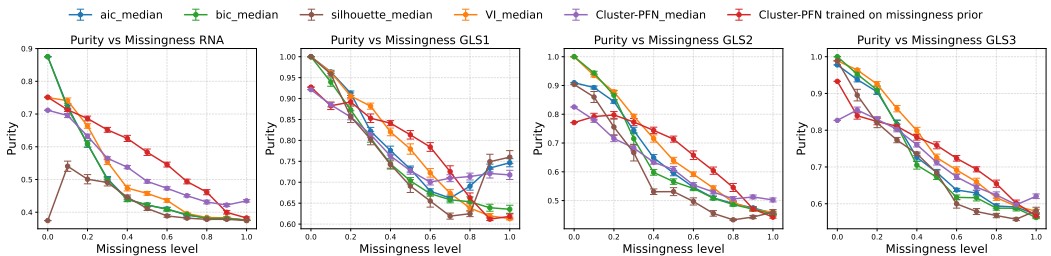

Figure 18: Purity scores for different models at varying missingness levels in real-world median imputed datasets. Error bars show standard error across 20 simulations (higher is better).

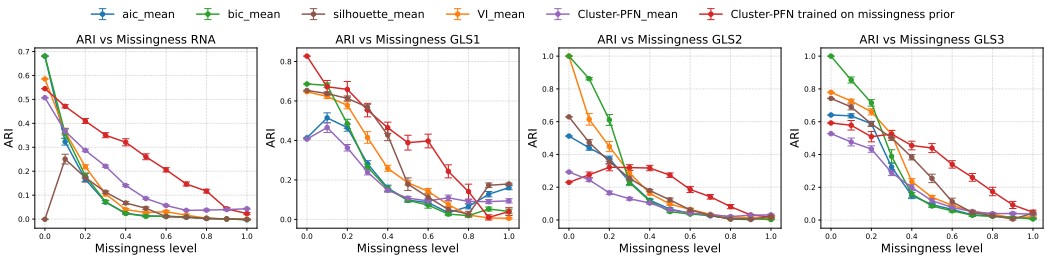

Figure 19: ARI scores for different models at varying missingness levels in real-world mean imputed datasets. Error bars show standard error across 20 simulations (higher is better).

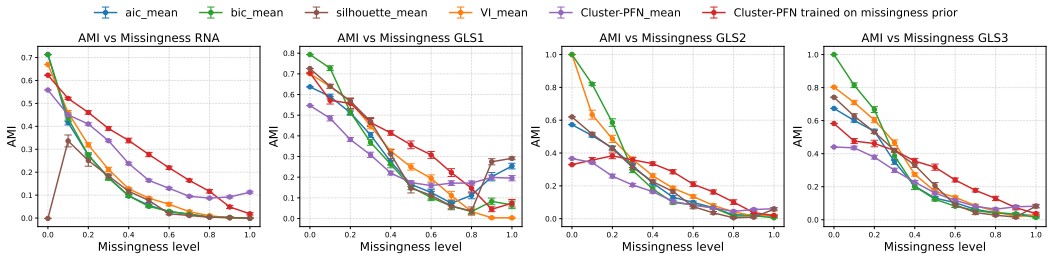

Figure 20: AMI scores for different models at varying missingness levels in real-world mean imputed datasets. Error bars show standard error across 20 simulations (higher is better).

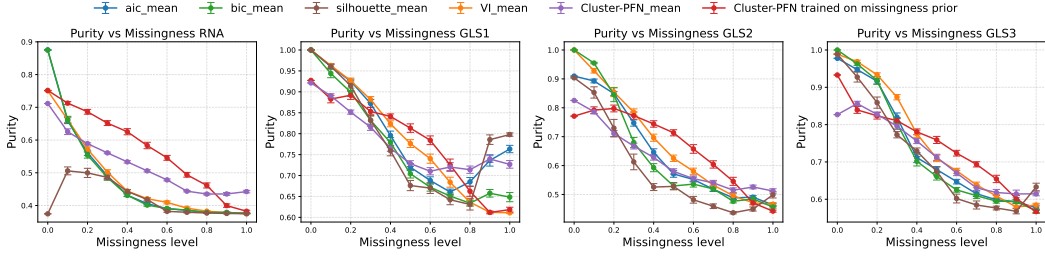

Figure 21: Purity scores for different models at varying missingness levels in real-world mean imputed datasets. Error bars show standard error across 20 simulations (higher is better).

