# OpenReview forum: "Transformers Can Do Bayesian Clustering"
_ICLR.cc/2026/Conference — Submitted to ICLR 2026_

### Official Review · Reviewer_Qm5R · 2025-10-18

**Soundness:** 1
**Presentation:** 3
**Contribution:** 1
**Rating:** 2
**Confidence:** 4

**Summary:**

The authors propose to use PFNs for Bayesian clustering. They introduce an augmented version of classical TabPFNs for the purpose of predicting the number of clusters and the cluster assignments in a given dataset.
They evaluate their approach on the synthetic training data sampled from their prior, and a limited set of real-world datasets and benchmark against a variational inference (VI) approach, as well as "GMM" utilizing different information criteria. Several ablations, including one on relatively large datasets with 10 000 datapoints is conducted.
Furthermore, a way to make the clustering robust to missing data is discussed and evaluated.

**Strengths:**

The paper is relatively well-written and follows a clear structure. While some important details on the baselines and datasets are missing, the most critical aspects of the experiments and the approach are made very clear. Them explicitly stating research questions helps the structure.

To my best knowledge, using PFNs to predict cluster assignments **together with** the number of clusters has not been explored. The idea to make the model also robust to missing data is nice!

In terms of experiments, the results they obtain look correct and reasonable and the large-dataset experiment (10,000 points) is interesting.

The findings overall are somewhat interesting from a conceptual point of view or could form the foundation for very specific applications of PFNs to clustering where all other methods fail.

**Weaknesses:**

Unfortunately, there are several major weaknesses:

1.) The paper completely ignores related work. Especially the paper "Reuter, Arik, et al. "Can Transformers Learn Full Bayesian Inference in Context?." Forty-second International Conference on Machine Learning." is highly relevant and seems to look into very related aspects of PFNs. In this paper, the authors also consider a PFN-approach to Bayesian clustering using GMMs and provide detailed experimental results. A thorough discussion of this paper is missing. Any other discussion of related work is also absent.

2.) The paper massively overclaims the scalability of the approach in the abstract. Not only do the authors not even consider datasets with more than five features, but also don't include any results on reasonably-sized **real-world** datasets. It is also highly questionable that proposed PFN approach (which do in-context learning) has any conceptual advantage in terms of scalability compared to other methods.

3.) Insufficient baselines: Just using one type of VI, that is not even properly explained or introduced, is clearly insufficient. The authors should explain and justify why they choose this particular type of VI and ensure that its hyperparamters are correctly set. Further VI baselines would also be needed for thorough experiments. Any sampling-based methods (MCMC) to perform inference are also missing.

4.) Insufficient real-world experiments: Only one real-world dataset for the non-missing scenario is definitely not enough and makes the reader highly suspicious.

5.) Insufficient number of tasks: It would be very interesting to see the Cluster PFN being trained on models other than GMMs to truly investigate Clustering as a problem and not just GMM clustering.

5.) Details, including implementation details, on the datasets and all baselines are missing.

6.) Lacking Novelty: The approach is conceptually very similar to existing PFN approaches and investigates essentially the same task as Reuter et al.

7.) The real-world applicability of this particular method is quite questionable. Why should anyone bother to fit a GMM with such an inefficient approach?

**Questions:**

Why do the authors believe that their method is scalable?

Why is the proposed PFN method conceptually a good approach for (Bayesian) clustering?

What exactly is the type of VI that is used?

Which hyperparameters are used for the VI method?

Why don't the author's consider other VI methods?

Why don't the authors consider sampling-based methods, in particular Hamiltonian Monte Carlo Samplers?

---

> ### Author Response · Authors · 2025-11-24
>
> We thank the reviewer for their feedback and for highlighting areas where our work can be improved. We will address their remarks and questions in detail below.
>
> ## W1: Regarding the paper Reuter, Arik, et al. "Can Transformers Learn Full Bayesian Inference in Context?
>
> *(the paper) is highly relevant and seems to look into very related aspects of PFNs. In this paper, the authors also consider a PFN-approach to Bayesian clustering using GMMs and provide detailed experimental results. A thorough discussion of this paper is missing. Any other discussion of related work is also absent.*
>
> Excellent remark! We in fact carried out this study in November 2024 - July 2025, which is why we must have missed this work. We had some related works; but in the end they were so far removed from our work that we have decided to remove them, and favored a “background section” instead. Thanks for bringing this paper to our attention! Indeed, it seems this paper is highly related, yet tackling a different question. Below, we describe the key differences between our work and Reuter, Arik, et al (2025)[1].
>
> **A different goal.** Reuter, Arik, et al. are developing a model to estimate the posterior distribution of the parameters, e.g. for the GMM case, they are estimating the distribution of the means, variances of the clusters. They assume the number of clusters is known and fixed, and do not aim to infer it. Instead, we aim to learn the number of clusters and we perform exact Bayesian inference over it. In comparison to their work, they are performing Full Bayesian inference over continuous variables and computing their distributions using Diffusion. We are estimating only discrete variables, e.g. the number of clusters and the responsibilities.
>
> This is why for our setting our training is different: we can simply train with cross-entropy, and we can evaluate the cluster numbers using accuracy. For evaluating clusterings, we use typical clustering metrics. Reuter, Arik, et al. only evaluate the quality of the distribution of their parameters, but do not evaluate their actual clusterings. In fact, they do not estimate the responsibilities at all, but only estimate for each datapoints the underlying cluster parameters.
>
> ***Our response is continued in the next comment.***
>
> ## References
> [1]Reuter, A., Rudner, T. G., Fortuin, V., & Rügamer, D. Can Transformers Learn Full Bayesian Inference in Context?. In Forty-second International Conference on Machine Learning.

---

> ### Author Response · Authors · 2025-11-24
>
> **Easier Clustering Setting**.  Reuter, Arik consider the following settings:
> - The number of clusters is either 3 or 5, and its always known how many clusters there are.
> - The number of datapoints of the dataset is always 25 or 50.
> - The dimensionality is either 1, 3 or 5; this is fixed and the model can only handle 1 dimensionality.
> - They consider a simple GMM prior, where the covariance matrix is diagonal, and thus only has 1-5 parameters (depending on the dimensionality) where the prior for the sigma’s on the diagonal are distributed according to an Inverse-Gamma prior. Each cluster has its own sigma’s.
>
> **Compare with our setting**:
> - The number of clusters is in [1,10]; it is always unknown how many clusters there are. We perform Full-Bayesian inference w.r.t. to the number of clusters.
> - The number of datapoints during training is in [100,500]. During evaluation we go up to 10k observations.
> - Our models can handle dimensionalities in [1,5]; our models are flexible and can deal with multiple dimensionalities.
> - We consider a more complex GMM prior. The covariance matrix of our clusters is always full ; e.g. for a 5D setting, each covariance matrix has 25 parameters. Each cluster has its own full covariance matrix (they are not tied). We use a standard Inverse-Wishart prior to sample the covariance matrices.
>
> **Remark regarding evaluation on real-world data.**  Reuter, Arik, et al. do not evaluate specifically on clustering datasets. Instead, they sample datasets for regression problems from OpenML based on  Grinsztajn et al. (2022)[2]. They pre-process the datasets in such a way that they are suitable for their models, e.g. for high-dimensional and large datasets, the data is subsampled to have less observations and less features; and in fact they thus evaluate also on maximally 5 dimensional data. What is of course problematic in clustering is that there is no-ground truth data. Grinsztajn et al. sidestep this issue, by performing an expensive Fully Bayesian computation using Markov-Chain Monte Carlo. The provided posterior samples are used for evaluation. This aligns with their goal, where they want to compare their posterior distribution with those of MCMC.
>
> This is not feasible for us, because running MCMC on GMM clustering is very difficult. We would like to cite the authors of Stan - the "go-to" package for Bayesian analysis. These authors that are experts in Bayesian inference write: “Two problems make it pretty much impossible to perform full Bayesian inference for clustering models, the lack of parameter identifiability and the extreme multimodality of the posteriors. There is additional discussion related to the non-identifiability due to label switching in the label switching section.” See https://mc-stan.org/docs/2_20/stan-users-guide/the-difficulty-of-bayesian-inference-for-clustering.html.
>
> This is likely why Reuter, Arik, et al. did not consider varying and estimating the number of clusters. For our real-world data, we have therefore stuck to clustering datasets that are accepted in the clustering community, where some groundtruth labels are known.
>
> **We want fast Bayesian GMM Clustering.** We have explicitly outlined in our work why we don’t estimate the posterior distributions of the clusterings: this is slow. This is in fact exactly what Reuter, Arik, et al. do. Indeed, in their Table 8 (appendix) we can observe that their method is much slower compared to ours: taking approximately 94 seconds, while we compute clusterings in 0.02 seconds, which is around 5000 times faster. This is because they require multiple forward passes to estimate distributions using diffusion, an issue which we avoid.
> Note that, this also clearly illustrates why we do not consider the MCMC baseline. They report in Table 8 that MCMC takes 240 seconds - for an easier setting where the number of clusters is known. Therefore, we don’t see the point of including MCMC in our work.
>
> **A final remark regarding VI-baselines.** It should be noted that Reuter, Arik, et al. do not consider a conjugate prior for their GMM. This will make inference using VI and MCMC more difficult; and may warrant more advanced VI and MCMC solvers. However, since our prior is conjugate, an efficient Mean-Field VI approximation exists for estimating the posteriors with VI. Therefore, our clustering experimental setting favors VI.
>
> ## References
>
> [1]Reuter, A., Rudner, T. G., Fortuin, V., & Rügamer, D. Can Transformers Learn Full Bayesian Inference in Context?. In Forty-second International Conference on Machine Learning.
>
> [2]Grinsztajn, L., Oyallon, E., and Varoquaux, G. Why do treebased models still outperform deep learning on typical tabular data? Advances in neural information processing systems, 35:507–520, 2022.

---

> ### Author Response · Authors · 2025-11-24
>
> ## W2: Regarding  not including any results on reasonably sized real-world datasets with reasonably sized feature sets.
>
> Why do you say there are no results included in “reasonably-sized real-world datasets”? Especially in the medical domain where these clustering approaches find application, datasets are often small. We would like to point out that the clustering datasets used have been recently used in clustering studies[1,2,3], and that Reuter, Arik, et al.[4] also only considers datasets up to 5 dimensions.
>
> ## W3: Insufficient baselines and Unexplained VI
>
> We are sorry for the oversight, indeed a detailed description of our VI baseline appears to be missing. Please find it below.
>
> We consider a favorable setting for VI since our priors are conjugate. Assume (for the moment) the number of clusters is known. We consider the Mean-Field approximation for VI, which is natural in our setting of the GMM. The meanfield approximation means that the posterior distribution factorizes; e.g. we assume Q(\theta,z) = Q(\theta)Q(z), where we indicate Q as the approximating distribution of the posterior. In this case, the EM-algorithm which is normally used for solving regular GMMs, can now be directly applied to the Bayesian-GMM. The E-step and M-step can in this case be easily computed in closed form (analytically), please for additional details see Bishop Chapter 10.2.
>
> We perform model selection over the number of $k$ using the ELBO, which can also be easily computed from the VI approximation. Because this algorithm can suffer from multiple local optima, we rerun VI with multiple initializations, as suggested by Bishop. We use the Scikit-implementation of Bayesian GMM to carry out our experiments, as it provides a robust implementation, see here:https://scikit-learn.org/stable/modules/generated/sklearn.mixture.BayesianGaussianMixture.html. Since we can rely on the EM-type updates, this VI implementation does not require a learning rate.
>
> Because we believe that this VI implementation should be very strong for this setting (e.g. it only requires a rather weak Mean-Field assumption), we think it is not warranted to include other VI-baselines. Furthermore, MCMC baselines will be very slow, which is why we don’t think they warrant inclusion here. However, if the reviewers have specific VI or MCMC algorithms in mind that are known to perform well for GMM clustering - please let us know, we would be very interested to know these and include them in our paper.
>
> ## W4: Insufficient real world experiments and number of tasks
>
> We appreciate this suggestion, but we do not have sufficient time to address this during the rebuttal phase.
>
> ## References
>
> [1] Hoadley, K. A., Yau, C., Wolf, D. M., Cherniack, A. D., Tamborero, D., Ng, S., Leiserson, M. D. M., Niu, B., McLellan, M. D., Uzunangelov, V., Zhang, J., Kandoth, C., Akbani, R.,
> Shen, H., Omberg, L., Chu, A., Margolin, A. A., Van't Veer, L. J., Lopez-Bigas, N., Laird, P. W., … Stuart, J. M. (2014). Multiplatform analysis of 12 cancer types reveals molecular classification within and across tissues of origin. Cell, 158(4), 929–944.
>
> [2] Taskesen, E., Huisman, S. M., Mahfouz, A., Krijthe, J. H., De Ridder, J., Van De Stolpe, A., ... & Reinders, M. J. (2016). Pan-cancer subtyping in a 2D-map shows substructures that are driven by specific combinations of molecular characteristics. Scientific reports, 6(1), 24949.
>
> [3] Julienne, H., Laville, V., McCaw, Z. R., He, Z., Guillemot, V., Lasry, C., ... & Aschard, H. (2021). Multitrait GWAS to connect disease variants and biological mechanisms. PLoS genetics, 17(8), e1009713.
>
> [4]Reuter, A., Rudner, T. G., Fortuin, V., & Rügamer, D. Can Transformers Learn Full Bayesian Inference in Context?. In Forty-second International Conference on Machine Learning.

---

> ### Author Response · Authors · 2025-11-24
>
> ## W5: Implementation and dataset details missing
>
> VI: We use the Scikit BayesianGuassianMixture class:https://scikit-learn.org/stable/modules/generated/sklearn.mixture.BayesianGaussianMixture.html. We set hyperparameters across all experiments to: Covariance_type = ‘full’, tol = 0.001, max_iter=200, n_init = 10, weight_concentration_prior_type  = ‘dirichlet_distribution’, weight_concentration_prior = 1, mean_prior = 0, degrees_of_freedom = # of features in data.The rest are set to the default.  As we have Easy and Hard datasets generated from the synthetic data generation, we set the mean_precision_prior (β) hyperparameter to 0.01, and 0.1 to the Easy and Hard datasets respectively. For the real world data, we choose to use mean_precision_prior of 0.1 as it performed better than 0.01. For model selection, we iterate over clusters 2 to $K = 10$ fitting the VI model and take the fit with the lowest ELBO.
>
> GMM: We use the Scikit GMM class: https://scikit-learn.org/stable/modules/generated/sklearn.mixture.GaussianMixture.html. We set covariance_type = ‘full’, max_iter = 200, n_init = 10, init_params = ‘kmeans’ and the rest are set to the default.
>
> We use several real-world classification datasets that have found use in studying GMMs. We chose these datasets as they have been used in the work by McCaw et al. (2022)[3] who develop a novel approach to fitting GMM models to missing data.
>
> **RNA dataset [1]**: The RNA dataset contains expression values for 20,531 genes across 801 patient samples from five tumor types. PCA is applied to reduce the data to five dimensions, following McCaw et al. (2022).
>
> **GWAS datasets [2]**: The GWAS provides a set of real summary statistics for cardiovascular disease with specific traits such as body mass index (BMI), coronary artery disease (CAD),low density lipoprotein (LDL), triglycerides (TG), waist to hip ratio (WHR), and any strokes (AS)
>
> | **Dataset** | **# of Classes** | **# of Features** | **# of Data Points** | **Description**
> | ------ | ---------- | -------- | -------------------- |----------------
> | **RNA**     | 5  | 5| 801 |
> | **GLS1**    | 3 | 3 | 165 | Feature types: BMI, CAD, LDL
> | **GLS2**    | 3 | 5 | 166 |Feature types:LDL, TG, BMI, AS, CAD
> | **GLS3**    | 3 | 5 | 179 | Feature types: LDL, TG, BMI, AS, WHR
>
> ## W7: real world applicability
>
> *Why should anyone bother to fit a GMM with such an inefficient approach?*
>
> Actually, our method is highly efficient, because after one upfront cost, the model (which we will publish) can be used to fit GMMs very quickly.
>
> ## References
> [1]Weinstein, J. N., Collisson, E. A., Mills, G. B., Shaw, K. R., Ozenberger, B. A., Ellrott, K., … & Stuart, J. M. (2013). The cancer genome atlas pan-cancer analysis project. Nature genetics, 45(10), 1113-1120.
>
> [2]Julienne, H., Lechat, P., Guillemot, V., Lasry, C., Yao, C., Araud, R., ... & Aschard, H. (2020). JASS: command line and web interface for the joint analysis of GWAS results. NAR genomics and bioinformatics, 2(1), lqaa003.
>
> [3]McCaw, Z.R., Aschard, H. & Julienne, H. Fitting Gaussian mixture models on incomplete data. BMC Bioinformatics 23, 208 (2022)

---

### Official Review · Reviewer_qNDq · 2025-10-28

**Soundness:** 1
**Presentation:** 1
**Contribution:** 2
**Rating:** 2
**Confidence:** 4

**Summary:**

The paper introduces Cluster-PFN, a Transformer-based model for computing posterior cluster assignments.
The model is trained entirely on synthetic Gaussian mixture datasets to predict both the number of clusters and posterior cluster responsibilities in a single (or two-step) forward pass.

**Strengths:**

1) The idea of adapting PFNs to perform Bayesian clustering is interesting and original.

2) The code for reproducibility is available.

3) The authors also discuss the limitations of the proposed approach.

**Weaknesses:**

See *Questions*.

**Questions:**

1) The paper is very difficult to follow. Already from the abstract, key acronyms (AIC, BIC) appear before being defined. Throughout the text, the authors make strong but insufficiently supported claims (“The results are clear”, “Cluster-PFN approximates the true Bayesian posterior over the number of clusters”), yet the actual mechanism of the model remains opaque. After several readings, it is still unclear what Cluster-PFN concretely does and how it differs from the seminal work of Muller et al. In practice, the model appears to be trained via supervised meta-learning on synthetic GMMs and to only imitate posterior-like outputs, without estimating latent parameters or uncertainty over model parameters. **The work needs substantial rewriting**.

2) The presentation of results is confusing, with synthetic and real-data experiments mixed together and no clear boundary between what is meant to demonstrate “proof of concept” and what supports the claimed generalization ability. Most importantly, the Discussion section exposes a conceptual contradiction. The entire point of PFN-like and meta-learning models is to perform meta-training on synthetic tasks so that the learned model can generalize to a wide variety of real-world datasets. However, the authors themselves acknowledge that their approach must be trained “for a particular prior” and that “Cluster-PFN is not always competitive on real-world data, only offering clear benefits on the GLS1 dataset.” This statement undermines the main motivation of the work: if the model needs to be retrained for each prior and fails to generalize across domains, it is unclear what advantage Cluster-PFN offers over standard inference methods.

3) The model formulation is difficult to follow. It is not clear how the conditioning on the number of clusters $k$ is implemented in practice, and why two forward passes are required when $k=0$. This seems to contradict the usual “single forward-pass inference” property of PFNs.
- Could the authors clarify how these two stages (estimating $k$ and computing cluster responsibilities) interact in the final inference pipeline?
- In addition, the model breaks label permutation invariance by assigning cluster labels through a fixed heuristic (“the cluster closest to the origin is label 0”). How much does this arbitrary rule influence the learned mapping?

---

> ### Author Response · Authors · 2025-11-24
>
> We thank the reviewer for their feedback and for highlighting areas where our work can be improved. We will address their remarks and questions in detail below.
>
> ## Cluster-PFN remains unclear
>
> We are sad to hear that you found our work hard to read. Is it possible for you to make constructive suggestions on what we can improve? Note that in your description “the model appears to be trained via supervised meta-learning on synthetic GMMs and to only imitate posterior-like outputs, without estimating latent parameters or uncertainty over model parameters.” appears accurate. “To only imitate” seems to point to a weakness of our approach (?), but we would like to remind the reviewer that the seminal work of Müller et al[1]. is in fact also trained in this way. We would really appreciate constructive feedback here to further improve our work, we are not sure what to change based on this question.
>
> ## Confusing presentation results
>
> *Cluster-PFN is not always competitive on real-world data, only offering clear benefits on the GLS1 dataset.” This statement undermines the main motivation of the work.*
>
> We are sorry, our statement was too strong: “Cluster-PFN is not always competitive on real-world data, only offering clear benefits on the GLS1 dataset.”. Clearly, the Cluster-PFN offers benefits especially in the case of missing data; for most values of missingness actually the Cluster-PFN beats the baselines significantly. Only in the zero-missingness case the Cluster-PFN is not always competitive, only offering a clear advantage on GLS1 in this setting.
>
> ## Model formulation
>
> *Could the authors clarify how these two stages (estimating  and computing cluster responsibilities) interact in the final inference pipeline?*
>
> In the final pipeline, first a forward pass is used to get the histogram $p(k|X)$. Afterward, we select the cluster with the highest posterior from this histogram (or the user selects whichever $k$ they prefer based on the histogram), which is used as input $k$ to the model again to obtain the responsibilities. This means that we require 2 forward-passes instead of one. One forward pass to determine the posterior over $k$, and one forward pass to estimate responsibilities conditioned on a (user specified) $k$.
>
>
> *In addition, the model breaks label permutation invariance by assigning cluster labels through a fixed heuristic (“the cluster closest to the origin is label 0”). How much does this arbitrary rule influence the learned mapping?*
>
> Note that this practice of breaking the labeling permutations is also applied in typical Bayesian modeling, see here:https://mc-stan.org/docs/2_20/stan-users-guide/label-switching-problematic-section.html#label-switching-problematic.section where the authors of Stan suggest this “trick” to make Bayesian inference using MCMC more well-behaved.
>
> We are not sure what is the influence of this arbitrary heuristic and unfortenately do not have the time during this rebuttal period to investigate this.
>
> ## References
>
> [1] Transformers Can Do Bayesian Inference. Müller, S., Hollmann, N., Pineda Arango, S., Grabocka, J., & Hutter, F. In: Proceedings of the 10th International Conference on Learning Representations (ICLR 2022).

---

### Official Review · Reviewer_y1XC · 2025-10-31

**Soundness:** 1
**Presentation:** 3
**Contribution:** 1
**Rating:** 2
**Confidence:** 4

**Summary:**

The paper presents a modification of Prior-Data Fitted Networks (PFNs) to implicitly perform Bayesian clustering. The model is trained on synthetic datasets generated from Gaussian Mixture Models (GMMs) with specified priors, using the corresponding cluster assignments as supervision for a Transformer. To support inference of the number of clusters, the model is also provided with special tokens that guide the prediction of cluster counts. Empirical evaluation focuses primarily on synthetic data generated from GMMs with dimensionality up to five, with a limited number of real-world datasets included for additional validation.

**Strengths:**

- The paper reads well overall, making it easy for the reader to follow the core ideas.

- To the best of my knowledge, this is the first work to apply Prior-Data Fitted Networks (PFNs) to clustering, opening a promising new direction for amortized Bayesian inference in unsupervised learning.

**Weaknesses:**

- **Limited methodological novelty**: the approach primarily adapts existing PFNs by treating known cluster assignments as supervised labels, with only minor modifications to the training procedure. This incremental extension reduces the overall contribution, and in my opinion, could only be mitigated if highly significant results were provided, which is not the case.

- **Lack of motivation**: while the abstract emphasizes *"missingness"* as a central motivation, this aspect is not meaningfully developed in the main text. It is only briefly addressed in the experiments through a simple masking setup, making it feel peripheral and disconnected from the paper's core contributions.

- **Overly simplistic datasets**: unless supported by references to recent work, the use of five-dimensional synthetic data generated from Gaussian mixtures appears insufficient for evaluating clustering performance. This setting does not reflect the complexity of modern clustering tasks, which often involve high-dimensional, noisy, and structurally diverse data.

- **Weak baselines**: similarly, the baseline methods used (e.g., GMM and variational inference) are outdated and overly simplistic, limiting the relevance of the empirical comparisons to current challenges in clustering.

- **Weak empirical evaluation**: the evaluation relies heavily on qualitative visualizations using simple, easily separable datasets. This is insufficient for demonstrating the robustness or scalability of the approach.

- **Unjustified computational cost**: while inference is reported to be ~50× faster, the training cost is cited as 60 GPU hours for clustering on five-dimensional Gaussian blobs, which seems unreasonable given the simplicity of the task, and reduces the practicality of the method.

- **Lack of experimental transparency**: key details are missing. I include some of them in the questions section.

### Minor Issues

- The methodological section (Section 3) is too brief and lacks details. Several important methodological details are deferred to Section 4, which is nominally focused on experiments. This separation impairs the paper's readability and logical flow.

- The use of the term *"Bayesian prior"* to describe data generated from a GMM does not align with the conventional meaning of priors in Bayesian inference—specifically, priors over model parameters or cluster assignments. This creates confusion.

- The Old Faithful dataset is not mentioned in the real-world dataset descriptions, yet it appears in the experimental results. This inconsistency should be addressed.

**Questions:**

- How is the maximum number of generated clusters $𝐾$ defined?

- How does the $\beta$ parameter produce cluster overlapping?

- What are the hyperparameters and implementation details for the baselines?

- The real-world datasets are poorly described, limiting reproducibility and interpretability.

---

> ### Author Response · Authors · 2025-11-24
>
> We thank the reviewer for their thoughtful feedback and appreciate their point about the missingness not being emphasized as part of our central motivation. We address their remarks and questions in detail below.
>
> ## Overly simplistic datasets
>
> We do not understand the remark regarding overly simplistic datasets. The reviewer writes: “unless supported by references to recent work”? The real-world datasets that we use are in fact supported by recent work in GMM Clustering papers. See line 298-303 of our paper: “To evaluate clustering under missingness, we use four real-world genomic benchmark datasets: one RNA sequencing dataset (Network et al., 2013)[1] and three GWAS summary statistics datasets (Julienne et al., 2020)[2], assessing external metric scores. These datasets were also used in prior work by McCaw et al.(2022)[3] to study missingness, which motivated our choice, though that study applied a standard GMM rather than a Bayesian approach.”.
>
> Furthermore in the medical domain where these clustering approaches find application, datasets are often small. We would like to point out that the clustering datasets used have been recently used in clustering studies[4,5,6], and that Reuter, Arik, et al. (2025)[7] (a state-of-the-art reference) also only considers datasets up to 5 dimensions.
>
> ## Weak baselines
>
> *methods used (e.g., GMM and variational inference) are outdated and overly simplistic, limiting the relevance of the empirical comparisons to current challenges in clustering.*
>
> If GMM is outdated, why would  McCaw et al. (2022)[3] study its properties under missingness, and develop a package for GMMs to handle missingness? Note that their work deals with vanilla GMM’s - not Bayesian ones that we consider. In fact, we believe Bayesian GMMs are rather exotic - we found it hard to find any prior work that benchmarks Bayesian GMMs for clustering. Could you please provide any references of recent work that we have missed?
>
> ## References
> [1]Weinstein, J. N., Collisson, E. A., Mills, G. B., Shaw, K. R., Ozenberger, B. A., Ellrott, K., … & Stuart, J. M. (2013). The cancer genome atlas pan-cancer analysis project. Nature genetics, 45(10), 1113-1120.
>
> [2]Julienne, H., Lechat, P., Guillemot, V., Lasry, C., Yao, C., Araud, R., ... & Aschard, H. (2020). JASS: command line and web interface for the joint analysis of GWAS results. NAR genomics and bioinformatics, 2(1), lqaa003.
>
> [3]McCaw, Z.R., Aschard, H. & Julienne, H. Fitting Gaussian mixture models on incomplete data. BMC Bioinformatics 23, 208 (2022)
>
> [4] Hoadley, K. A., Yau, C., Wolf, D. M., Cherniack, A. D., Tamborero, D., Ng, S., Leiserson, M. D. M., Niu, B., McLellan, M. D., Uzunangelov, V., Zhang, J., Kandoth, C., Akbani, R.,
> Shen, H., Omberg, L., Chu, A., Margolin, A. A., Van't Veer, L. J., Lopez-Bigas, N., Laird, P. W., … Stuart, J. M. (2014). Multiplatform analysis of 12 cancer types reveals molecular classification within and across tissues of origin. Cell, 158(4), 929–944.
>
> [5] Taskesen, E., Huisman, S. M., Mahfouz, A., Krijthe, J. H., De Ridder, J., Van De Stolpe, A., ... & Reinders, M. J. (2016). Pan-cancer subtyping in a 2D-map shows substructures that are driven by specific combinations of molecular characteristics. Scientific reports, 6(1), 24949.
>
> [6] Julienne, H., Laville, V., McCaw, Z. R., He, Z., Guillemot, V., Lasry, C., ... & Aschard, H. (2021). Multitrait GWAS to connect disease variants and biological mechanisms. PLoS genetics, 17(8), e1009713.
>
> [7]Reuter, A., Rudner, T. G., Fortuin, V., & Rügamer, D. Can Transformers Learn Full Bayesian Inference in Context?. In Forty-second International Conference on Machine Learning.
>
> [8] Transformers Can Do Bayesian Inference. Müller, S., Hollmann, N., Pineda Arango, S., Grabocka, J., & Hutter, F. In: Proceedings of the 10th International Conference on Learning Representations (ICLR 2022).
>
> [9] Accurate Predictions on Small Data with a Tabular Foundation Model. Hollmann, N., Müller, S., Purucker, L., Krishnakumar, A., Körfer, M., Hoo, S. B., Schirrmeister, R. T., & Hutter, F. Published in Nature (2025).
>
> [10] Adriaensen, S., Rakotoarison, H., Müller, S., & Hutter, F. (2023). Efficient Bayesian learning curve extrapolation using prior-data fitted networks. Advances in Neural Information Processing Systems, 36, 19858-19886.
>
> [11] Müller, S., Feurer, M., Hollmann, N., & Hutter, F. (2023, July). Pfns4bo: In-context learning for bayesian optimization. In International Conference on Machine Learning (pp. 25444-25470). PMLR.

---

> ### Author Response · Authors · 2025-11-24
>
> ## Weak empirical evaluation
>
> *the evaluation relies heavily on qualitative visualizations using simple, easily separable datasets*
>
> We think this reviewer's comment is misleading or perhaps some crucial results were missed; given that we provide quantitative results; see Table 1, Figure 5, and Figure 6. Furthermore, we evaluate both on EASY and HARD datasets; therefore we would strongly disagree that we “evaluate only using simple easily separable datasets”.  We added an example of an easy and hard dataset on page 10 of the manuscript to more clearly illustrate them.
>
> ## Unjustified computational cost
>
> *While inference is 50x faster, training cost is cited as 60 GPU hours for clustering on five-dimensional Gaussian blobs, which seems unreasonable given the simplicity of the task*
>
> We would like to cite the authors of Stan - the go-to package for Bayesian analysis. These authors, who are experts in Bayesian inference, write: “Two problems make it pretty much impossible to perform full Bayesian inference for clustering models: the lack of parameter identifiability and the extreme multimodality of the posteriors. There is additional discussion related to the non-identifiability due to label switching in the label switching section.” See https://mc-stan.org/docs/2_20/stan-users-guide/the-difficulty-of-bayesian-inference-for-clustering.html. This should make it evident that Fully Bayesian Clustering is far from trivial. Furthermore, PFNs often require very long training times. For example, the PFN from Müller et al. (2022) [8] requires several days to train, the TabPFN from Hollmann et al.(2025) [9] reportedly takes two weeks on eight GPUs, and the LC-PFN from Adriaensen et al. (2022)[10] requires around eight hours.
>
> ## Term "Bayesian prior" to describe data generated from a GMM does not align with the conventional meaning
>
> Actually, we would refer to at least PFN4BO [11] which also mentions the use of different priors. E.g. a GP-prior, BNN-prior, etc. This is also natural, especially in cases with hierarchical Bayesian modeling, where multiple different models (e.g. priors) could explain the data.
>
> ## How is the maximum number of generated clusters  defined?
>
> Sorry, it seems we have forgotten to mention this. We fix K=10 maximally everywhere.
>
>
> ## References
> [1]Weinstein, J. N., Collisson, E. A., Mills, G. B., Shaw, K. R., Ozenberger, B. A., Ellrott, K., … & Stuart, J. M. (2013). The cancer genome atlas pan-cancer analysis project. Nature genetics, 45(10), 1113-1120.
>
> [2]Julienne, H., Lechat, P., Guillemot, V., Lasry, C., Yao, C., Araud, R., ... & Aschard, H. (2020). JASS: command line and web interface for the joint analysis of GWAS results. NAR genomics and bioinformatics, 2(1), lqaa003.
>
> [3]McCaw, Z.R., Aschard, H. & Julienne, H. Fitting Gaussian mixture models on incomplete data. BMC Bioinformatics 23, 208 (2022)
>
> [4] Hoadley, K. A., Yau, C., Wolf, D. M., Cherniack, A. D., Tamborero, D., Ng, S., Leiserson, M. D. M., Niu, B., McLellan, M. D., Uzunangelov, V., Zhang, J., Kandoth, C., Akbani, R.,
> Shen, H., Omberg, L., Chu, A., Margolin, A. A., Van't Veer, L. J., Lopez-Bigas, N., Laird, P. W., … Stuart, J. M. (2014). Multiplatform analysis of 12 cancer types reveals molecular classification within and across tissues of origin. Cell, 158(4), 929–944.
>
> [5] Taskesen, E., Huisman, S. M., Mahfouz, A., Krijthe, J. H., De Ridder, J., Van De Stolpe, A., ... & Reinders, M. J. (2016). Pan-cancer subtyping in a 2D-map shows substructures that are driven by specific combinations of molecular characteristics. Scientific reports, 6(1), 24949.
>
> [6] Julienne, H., Laville, V., McCaw, Z. R., He, Z., Guillemot, V., Lasry, C., ... & Aschard, H. (2021). Multitrait GWAS to connect disease variants and biological mechanisms. PLoS genetics, 17(8), e1009713.
>
> [7]Reuter, A., Rudner, T. G., Fortuin, V., & Rügamer, D. Can Transformers Learn Full Bayesian Inference in Context?. In Forty-second International Conference on Machine Learning.
>
> [8] Transformers Can Do Bayesian Inference. Müller, S., Hollmann, N., Pineda Arango, S., Grabocka, J., & Hutter, F. In: Proceedings of the 10th International Conference on Learning Representations (ICLR 2022).
>
> [9] Accurate Predictions on Small Data with a Tabular Foundation Model. Hollmann, N., Müller, S., Purucker, L., Krishnakumar, A., Körfer, M., Hoo, S. B., Schirrmeister, R. T., & Hutter, F. Published in Nature (2025).
>
> [10] Adriaensen, S., Rakotoarison, H., Müller, S., & Hutter, F. (2023). Efficient Bayesian learning curve extrapolation using prior-data fitted networks. Advances in Neural Information Processing Systems, 36, 19858-19886.
>
> [11] Müller, S., Feurer, M., Hollmann, N., & Hutter, F. (2023, July). Pfns4bo: In-context learning for bayesian optimization. In International Conference on Machine Learning (pp. 25444-25470). PMLR.

---

> ### Author Response · Authors · 2025-11-24
>
> ## How does the β parameter produce cluster overlapping?
> The cluster means μᵢ are sampled from a Gaussian distribution whose covariance is $Σᵢ/β$ (Σ is sampled the same way; irrespective of β). The smaller β is, the more widely the cluster means are spread; the larger β is, the more tightly they cluster around the origin, producing overlap. This way,  β makes either clusters overlap more or less. We added a figure to explain this qualitatively  which can be seen on page 10 of the manuscript.
>
> ## What are the hyperparameters and implementation details for the baselines?
>
> VI: We use the Scikit BayesianGuassianMixture class:https://scikit-learn.org/stable/modules/generated/sklearn.mixture.BayesianGaussianMixture.html. We set hyperparameters across all experiments to: Covariance_type = ‘full’, tol = 0.001, max_iter=200, n_init = 10, weight_concentration_prior_type  = ‘dirichlet_distribution’, weight_concentration_prior = 1, mean_prior = 0, degrees_of_freedom = # of features in data.The rest are set to the default.  As we have Easy and Hard datasets generated from the synthetic data generation, we set the mean_precision_prior (β) hyperparameter to 0.01, and 0.1 to the Easy and Hard datasets respectively. For the real world data, we choose to use mean_precision_prior of 0.1 as it performed better than 0.01. For model selection, we iterate over clusters 2 to $K = 10$ fitting the VI model and take the fit with the lowest ELBO.
>
> GMM: We use the Scikit GMM class: https://scikit-learn.org/stable/modules/generated/sklearn.mixture.GaussianMixture.html. We set covariance_type = ‘full’, max_iter = 200, n_init = 10, init_params = ‘kmeans’ and the rest are set to the default.
>
> We use several real-world classification datasets that have found use in studying GMMs. We chose these datasets as they have been used in the work by McCaw et al. (2022)[3] who develop a novel approach to fitting GMM models to missing data.
>
> **RNA dataset [1]**: The RNA dataset contains expression values for 20,531 genes across 801 patient samples from five tumor types. PCA is applied to reduce the data to five dimensions, following McCaw et al. (2022).
>
> **GWAS datasets [2]**: The GWAS datasets (GLS-3) provide a set of real summary statistics for cardiovascular disease with specific traits such as body mass index (BMI), coronary artery disease (CAD),low density lipoprotein (LDL), triglycerides (TG), waist to hip ratio (WHR), and any strokes (AS)
>
> | **Dataset** | **# of Classes** | **# of Features** | **# of Data Points** | **Description**
> | ------ | ---------- | -------- | -------------------- |----------------
> | **RNA**     | 5  | 5| 801 |
> | **GLS1**    | 3 | 3 | 165 | Feature types: BMI, CAD, LDL
> | **GLS2**    | 3 | 5 | 166 |Feature types:LDL, TG, BMI, AS, CAD
> | **GLS3**    | 3 | 5 | 179 | Feature types: LDL, TG, BMI, AS, WHR
>
> ## References
> [1]Weinstein, J. N., Collisson, E. A., Mills, G. B., Shaw, K. R., Ozenberger, B. A., Ellrott, K., … & Stuart, J. M. (2013). The cancer genome atlas pan-cancer analysis project. Nature genetics, 45(10), 1113-1120.
>
> [2]Julienne, H., Lechat, P., Guillemot, V., Lasry, C., Yao, C., Araud, R., ... & Aschard, H. (2020). JASS: command line and web interface for the joint analysis of GWAS results. NAR genomics and bioinformatics, 2(1), lqaa003.
>
> [3]McCaw, Z.R., Aschard, H. & Julienne, H. Fitting Gaussian mixture models on incomplete data. BMC Bioinformatics 23, 208 (2022)

---

> ### Comment · Reviewer_y1XC · 2025-11-26
>
> I appreciate the authors’ effort in preparing the rebuttal. However, after reading the responses carefully, I find that the core concerns raised in my initial review remain largely unresolved. In particular, the issues related to baseline adequacy, the limited scope of the evaluation, and the disconnect between the stated motivation around missing data and the actual methodological contribution were not addressed in a substantive way.
>
> ## Baselines and related work
> Thank you for the additional clarification. I understand that the proposed method is designed specifically for Bayesian clustering under a finite GMM prior. My concern is not that the method should be evaluated on general clustering tasks, but that—even within this restricted GMM framework—the baselines used in the paper are not sufficiently representative of modern Bayesian mixture inference.
>
> McCaw et al. (2022) is not a methodological baseline but a Stan-based implementation of classical EM-style missing-data GMMs (Dempster et al., 1977; Ghahramani & Jordan, 1993). It does not introduce any new inference principle, and therefore does not justify treating vanilla GMMs, equipped with heuristic model-selection criteria (AIC, BIC, Silhouette), as strong comparators for a modern amortized Bayesian inference method.
>
> The rebuttal states that “Bayesian GMMs are exotic” and that it is “hard to find prior work” that benchmarks them. This is not accurate. There is a substantial body of empirical work evaluating Bayesian mixture models, including Dirichlet Process Mixtures (Dahl 2006), sparse Bayesian GMMs with extensive benchmarking (Yao et al. 2025), Bayesian fusion methods addressing overfragmentation (FOLD; Dombowsky & Dunson 2023), and ensemble-based Bayesian clustering approaches (Wang et al. 2011; Coleman et al. 2022). In addition, amortized deep Bayesian clustering frameworks—such as Neural Clustering Processes (Pakman et al., 2020), DeepDPM (Ronen et al., 2022), and AMCP (Liu et al., 2022)—include quantitative clustering evaluations highly relevant to the goals of the present work.
>
> These methods demonstrate that Bayesian GMM inference is neither rare nor lacking empirical comparisons. Limiting the baselines to AIC/BIC/Silhouette selection and mean-field VI therefore results in outdated and uninformative comparisons, and materially overstates the relative performance of Cluster-PFN.
>
> ## Data
> Although the method is derived under a finite GMM prior, the experiments rely almost exclusively on synthetic Gaussian mixtures generated directly from the same conjugate NIW prior used during training. Even the “hard” prior scenario corresponds to mildly overlapping Gaussians, remaining well within the model’s training distribution. This setup sidesteps many challenges that arise even within mixture models—such as heteroscedasticity, heavy tails, or deviations from ellipticity—that strongly affect Bayesian GMM inference.
>
> The existence of methods like FOLD (Dombowsky & Dunson 2023), which were developed to correct overfragmentation and instability of Bayesian GMMs on real data, underscores that real datasets rarely conform to simple Gaussian generative assumptions. While the paper does include a small set of real-data experiments, these evaluations remain very limited in scope and do not address the substantial prior–data mismatch induced by training exclusively on synthetic Gaussian mixtures. As a result, the conclusions drawn about generalization in RQ1 are not strongly supported by the current experimental design.
>
> ## Missing-data motivation
> I would also like to reiterate the concern regarding missing data. The abstract and introduction present missingness as a core motivation, citing the difficulty of Bayesian inference with partially observed inputs. However, the proposed approach is trained and evaluated almost entirely on fully observed synthetic datasets, with missingness introduced only artificially through simple random masking. There is no modeling of MCAR/MAR/MNAR mechanisms, no handling of missingness at the likelihood level, and no comparison to established Bayesian or deep generative models explicitly designed for missing-data inference. Consequently, the connection between the stated motivation and the empirical evidence remains weak.

---

> > ### Comment · Reviewer_y1XC · 2025-11-26
> >
> > ## References
> >
> > Dempster, A. P., Laird, N. M., & Rubin, D. B. (1977). Maximum likelihood from incomplete data via the EM algorithm. Journal of the royal statistical society: series B (methodological), 39(1), 1-22.
> >
> > Ghahramani, Z., & Jordan, M. (1993). Supervised learning from incomplete data via an EM approach. Advances in neural information processing systems, 6.
> >
> > Dahl, D. B. (2006). Model-based clustering for expression data via a Dirichlet process mixture model. Bayesian inference for gene expression and proteomics, 4, 201-218.
> >
> > Yao, D., Xie, F., & Xu, Y. (2025). Bayesian Sparse Gaussian Mixture Model for Clustering in High Dimensions. Journal of Machine Learning Research, 26(21), 1-50.
> >
> > Dombowsky, A., & Dunson, D. B. (2023). Bayesian Clustering via Fusing of Localized Densities. arXiv preprint arXiv:2304.00074.
> >
> > Wang, H., Shan, H., & Banerjee, A. (2011). Bayesian cluster ensembles. Statistical Analysis and Data Mining: The ASA Data Science Journal, 4(1), 54-70.
> >
> > Coleman, S., Kirk, P. D., & Wallace, C. (2022). Consensus clustering for Bayesian mixture models. BMC bioinformatics, 23(1), 290.
> >
> > Pakman, A., Wang, Y., Mitelut, C., Lee, J., & Paninski, L. (2020, November). Neural clustering processes. In International Conference on Machine Learning (pp. 7455-7465). PMLR.
> >
> > Ronen, M., Finder, S. E., & Freifeld, O. (2022). Deepdpm: Deep clustering with an unknown number of clusters. In Proceedings of the IEEE/CVF Conference on Computer Vision and Pattern Recognition (pp. 9861-9870).
> >
> > Liu, H., & Jing, L. (2022). Amortized mixing coupling processes for clustering. Advances in Neural Information Processing Systems, 35, 11714-11725.

---

### Official Review · Reviewer_3zPn · 2025-11-01

**Soundness:** 2
**Presentation:** 3
**Contribution:** 3
**Rating:** 4
**Confidence:** 4

**Summary:**

The authors extend Prior‑Data Fitted Networks (PFNs) from supervised prediction to *Bayesian clustering*. Their **Cluster‑PFN** is a Transformer that, given a set of points (X) and an input (k), outputs (i) a distribution over the *number of clusters* (P(k \mid X)) (when fed (k=0)) and (ii) *responsibilities* (p(z_i=k\mid X,k)) for each point when a cluster count is provided. A special “collector” token (\rho) aggregates information to predict (P(k\mid X)) (Figure 2, p.3), and the model conditions on (k) via an embedding added to all tokens.
The model is trained entirely on synthetic datasets drawn from a *finite GMM* with Normal–Inverse‑Wishart priors; versions include 2D and up to 5D inputs as well as *random missingness* up to 80%. Architecture uses a 4‑layer, 4‑head encoder with 256‑dim embeddings (Appendix C).

**Strengths:**

Demonstrates that a single Transformer forward pass—trained purely on synthetic prior‑samples—can approximate *both* (P(k\mid X)) and responsibilities, a compelling extension of PFNs into unsupervised Bayesian modeling. The special (\rho) token for (k) prediction (Fig. 2) and conditioning mechanism are simple but elegant.  Clear runtime wins vs VI, even when VI uses multiple inits (Table 2), and scaling tests up to 20k points show consistent advantages (times reported on p.8).

**Weaknesses:**

For AIC/BIC/silhouette, the search over (k) *excludes (k=1)* “since the silhouette score is undefined for a single cluster” (p.5). But AIC and BIC are perfectly well‑defined at (k=1). Excluding (k=1) likely *penalizes* AIC/BIC whenever the truth is one cluster, inflating Cluster‑PFN’s relative accuracy in Table 1. A fair protocol would allow (k\in{1,\ldots,K}) for AIC/BIC and handle silhouette separately.

The paper argues Cluster‑PFN “approximates the true Bayesian posterior over the number of clusters” by analogy with supervised PFNs, but *responsibilities are learned independently* , not via a coherent joint posterior over ((\theta,z)). Moreover, when (k) is unknown, the method ultimately uses a *two‑pass MAP (k)* rather than integrating over (k) (the fully Bayesian option the model initially formulates), because of label‑ordering bias. This iweakens the Bayesian claim for responsibilities.

 The model sometimes fails to obey a user‑specified (k), especially when the instruction is “wildly different” from the data’s structure; the authors quantify this and show accuracy improves when conditioning near the unconditioned prediction. For downstream pipelines that *require* exactly (k) clusters, this is a limitation.

 The deterministic relabeling uses distance to the origin after zero‑one scaling. While consistent across training tasks, it encodes an *arbitrary geometry* (e.g., clusters near (\mathbf{0}) get low indices), is sensitive to min–max scaling quirks, and is not feature‑permutation invariant—an issue the authors also list as a limitation (Section 7). More principled label alignment (e.g., Hungarian matching to learned prototypes) or an equivariant architecture would be cleaner.

The paper itself notes that on real datasets without severe missingness, Cluster‑PFN is “not always competitive,” with clear wins mainly on GLS1 and under high missingness (Section 6). This underscores *prior mismatch*: training solely on finite‑GMM priors may not capture real data complexity (also discussed on p.9).  Further,  experiments cap at 5D and produce 10 logits (Appendix C). This raises questions about behavior in high‑D tabular domains (common in genomics) and for larger (K). The paper mentions feature‑permutation invariance and scaling to higher (d) as future work (Section 7).

Transformer’s (O(N^2)) attention remains; so  very large (N) regimes may still be challenging without sparse/linear attention variants.

The paper reports NLL of responsibilities (with label‑permutation minimization) and shows a histogram (Fig. 5c–d), but does not assess calibration of (P(k\mid X)) or of per‑point responsibilities (e.g., reliability diagrams, ECE). This matters if outputs are to be trusted as Bayesian probabilities.

**Questions:**

Fix the (k=1) baseline issue  Re‑run AIC/BIC with (k\in{1,\ldots,K}); report per‑(k) accuracy and overall accuracy marginalizing (k\sim U(1,K)).

Compare against EM/VI that marginalize missing features directly for GMMs, not only imputation‑based pipelines.

Reliability diagrams and proper scoring (e.g., Brier) for (P(k\mid X)); temperature scaling if needed. Include calibration for responsibilities.

 Train on broader priors (non‑Gaussian, skewed/cluster‑size imbalance, heavy‑tailed, anisotropic covariances) and evaluate on real data; quantify sensitivity to prior misspecification (Section 6 hints at this).

 Demonstrate feature‑permutation‑equivariant variants (as suggested in Section 7) and report how accuracy and runtime change.

---

> ### Author Response · Authors · 2025-11-24
>
> We thank the reviewer for their feedback and for highlighting areas where our work can be improved. We will address their remarks and questions in detail below.
>
> ## The exclusion of $k=1$ cluster
> We have excluded the one cluster scenario in the respective evaluations to make sure the protocol is fair across the methods. Sorry, this was not clear from the manuscript. We will add it in the experimental setup section.
>
> ## two‑pass MAP (k) rather than integrating over (k)
>
> *...when (k) is unknown, the method ultimately uses a two‑pass MAP (k) rather than integrating over (k) (the fully Bayesian option the model initially formulates), because of label‑ordering bias. This weakens the Bayesian claim for responsibilities.*
>
> Our approach uses a fully-Bayesian treatment over $k$. After selecting $k$; again our method is fully Bayesian; but we condition on the true number of clusters. So, in fact, we never use MAP. In fact, we believe perhaps we were a bit too careful in our wording. Responsibilities only make sense when  conditioned on $k$. For example, assume we are integrating out the responsibilities for $k=1$ and $k=2$. By definition, all responsibilities are 1 for $k=1$, because there is only one cluster. Now for the case $k=2$, it may be possible that there are two clusters that are different from the $k=1$ case. Why would it make sense to then average these responsibilities over $k$; as the clusters are clearly different?
>
> ## model sometimes fails to obey a user‑specified $k$
>
> We agree and acknowledge this limitation, which is thoroughly investigated in Appendix E.1. Note that, often when $k$ is set to a reasonable amount (e.g. set to either the true number of clusters +-1 or +- 2, our method does work well).
>
> ## Deterministic relabeling scheme
>
> We agree that this is a weakness in our current architecture, but we do not have sufficient time to address this during the rebuttal phase. Note that this practice of breaking the labeling permutations is also applied in typical Bayesian modeling, see here:
> https://mc-stan.org/docs/2_20/stan-users-guide/label-switching-problematic-section.html#label-switching-problematic.section
>
> Where the authors of Stan suggest this “trick” to make Bayesian inference using MCMC more well-behaved.
>
> ## transformer scaling
>
> *Transformer’s (O(N^2)) attention remains; so very large (N) regimes may still be challenging without sparse/linear attention variants.*
>
> Zeng et al.[1] introduce TabFlex, a method that scales PFN models through linear attention. We believe that the same principles can be applied to our own model. Therefore, we don't see why it's not possible to also let our Cluster-PFN scale to millions of points.
>
> ## Comparison against EM/VI that marginalize missing features directly for GMMs
>
> We have carried out this experiment. However, we find that when we use missingness directly in VI for GMM clustering that the performance for estimating the number of clusters is severely weakened. Please find the results below.
>
> | Cluster number | ELBO
> | ------------- | ------------------
> | 2 | -1116
> |  3| -1130
> |  4| -1138
> |  5| -1175
> |  6| -1169
> |  7| -1186
> |  8| -1192
> |  9| -1200
>
>
> The table shows the ELBO values versus the number of clusters for the VI model fitted to the RNA dataset (with 5 true labels) containing 10% missing values, used here for model selection. Higher ELBO values indicate a better fit. We observe a decreasing relationship between ELBO and the number of clusters; and as such, this method will tend to select only 2 clusters. Therefore the model is very weak in uncovering the true number of clusters. Very similar patterns are seen across all of the real-world datasets. Thus, we find that this VI-variant that integrates missingness directly, underperforms generally compared to VI variants that use imputation. We are trying to diagnose the issue, but we think this might not be solvable and may have to do with a failure mode of VI with missing data for GMMs.
>
>
> ## References
> [1] Zeng, Y., Dinh, T., Kang, W., & Mueller, A. C. TabFlex: Scaling Tabular Learning to Millions with Linear Attention. In Forty-second International Conference on Machine Learning.

---

> ### Author Response · Authors · 2025-11-24
>
> ## Reliability and calibration for responsibilities
>
> We have added evaluation in terms of ECE for the responsibilities. For the VI and GMM models, we run model selection with 10 initializations and do this 10 times and compute average ECE scores of cluster assignments with standard deviation. Note that we only have one result for the PFN, as it is deterministic. Please see the table below.
> | **Models**                 | **RNA**     | **GLS1**      | **GLS2**      | **GLS3**      |
> | -------------------------- | ----------- | ------------- | ------------- | ------------- |
> | **PFN trained on missingness** | 0.002       | 0.014         | 0.004         | 0.006         |
> | **PFN**                    | 0.014       | 0.009         | 0.002         | 0.002         |
> | **VI**                     | 0.04 ± 0.02 | 0.01 ± 0.005  | 0.004 ± 0.005 | 0.007 ± 0.002 |
> | **GMM AIC**                | 0.04 ± 0.02 | 0.008 ± 0.05  | 0.012 ± 0.008 | 0.012 ± 0.007 |
> | **GMM BIC**                | 0.06 ± 0.05 | 0.023 ± 0.01  | 0.0009 ± 0    | 0.009 ± 0.01  |
> | **GMM silhouette**         | 0.07 ± 0    | 0.012 ± 0.008 | 0.006 ± 0.006 | 0.01 ± 0.005  |
>
> In comparison with VI, observe that the PFN is often a bit better calibrated.
>
> ## Train on broader priors and feature‑permutation‑equivariant variants
>
> We appreciate this suggestion, but we do not have sufficient time to address this during the rebuttal phase.

---

### Meta-Review · Area_Chair_VYZQ · 2026-01-09

**Summary:**

The paper proposes a transformer-based model that extends Prior-Data Fitted Networks (PFNs) to Bayesian clustering with missing values, i.e., performing Bayesian inference on the number of clusters and the cluster responsibilities for each data point. The reviews found the extension of PFN to clustering to be interesting and novel, and appreciated that the authors discussed the proposed method's limitations. Some reviews found the paper read well (y1XC and Qm5R), while others found it difficult to follow (qNDq). Reviewers raised many major concerns about ignoring related work, overclaiming, insufficient baselines, simplistic datasets and lack of real-world experiments, weak empirical evaluation, unclear methodological and experimental details (cf. reviewer qNDq found the paper difficult to read), a mismatch between the missing data motivation and the limited complexity of the missing-data evaluation, and limited methodological novelty over PFN.

**Reviewer Concerns:**

The rebuttal provides additional clarification and details, but many core concerns remain unresolved, including the extension to more difficult datasets or real-world experiments, insufficient baselines, and the motivation for the missing data.

**Reviewer Scores:**

All the reviewers' scores will likely remain unchanged, as they all raised multiple substantive issues, and only a few appear to have been fully addressed.

---

### Decision · Program_Chairs · 2026-01-26

Reject